# β-arrestin1 promotes tauopathy by transducing GPCR signaling, disrupting microtubules and autophagy

Jung-AA Woo[1] , Yan Yan[1,2], Teresa R Kee[1,2], Sara Cazzaro[1,2], Kyle C McGill Percy[1], Xinming Wang[1], Tian Liu[1], Stephen B Liggett[3] , David E Kang[1,4]

G protein–coupled receptors (GPCRs) have been shown to play integral roles in Alzheimer's disease pathogenesis. However, it is unclear how diverse GPCRs similarly affect Aβ and tau pathogenesis. GPCRs share a common mechanism of action via the β-arrestin scaffolding signaling complexes, which not only serve to desensitize GPCRs by internalization, but also mediate multiple downstream signaling events. As signaling via the GPCRs, β2-adrenergic receptor (β2AR), and metabotropic glutamate receptor 2 (mGluR2) promotes hyperphosphorylation of tau, we hypothesized that β-arrestin1 represents a point of convergence for such pathogenic activities. Here, we report that β-arrestins are not only essential for β2AR and mGluR2-mediated increase in pathogenic tau but also show that β-arrestin1 levels are increased in brains of Frontotemporal lobar degeneration (FTLD-tau) patients. Increased β-arrestin1 in turn drives the accumulation of pathogenic tau, whereas reduced *ARRB1* alleviates tauopathy and rescues impaired synaptic plasticity and cognitive impairments in PS19 mice. Biochemical and cellular studies show that β-arrestin1 drives tauopathy by destabilizing microtubules and impeding p62/SQSTM1 autophagy flux by interfering with p62 body formation, which promotes pathogenic tau accumulation.

## Introduction

Alzheimer's disease (AD) is characterized by the presence of amyloid-β plaques and neurofibrillary tangles, which are aggregates of amyloid β (Aβ) and hyperphosphorylated tau, respectively, in the brains of affected individuals. Multiple G protein–coupled receptors (GPCRs) have been shown to play a somewhat ill-defined role in AD pathogenesis (Lee et al, 2004; Minkeviciene et al, 2004; Sun et al, 2005; Ni et al, 2006; Bakshi et al, 2008; Lee et al, 2009; Thathiah et al, 2009; AbdAlla et al, 2009a, 2009b; Alley et al, 2010; Dobarro et al, 2013; Luong & Nguyen, 2013; Wisely et al, 2014). However, it is unclear how

the reported effects of agonists or antagonists acting at a diverse array of GPCRs and their cognate signaling pathways converge to have effects on Aβ and tau pathogenesis. Most GPCRs, though, do interact with the β-arrestins, which in their monomeric forms bind to the agonist-occupied phosphorylated receptor and attenuate signaling by binding near the receptor–G protein interface. This serves as a mechanism for rapidly regulating pre- and postsynaptic receptor function. β-arrestins also initiate other, non–G protein mediated cellular events by creating signaling complexes because of their scaffolding actions (Wilden et al, 1986; Lohse et al, 1990; Gurevich & Gurevich, 2006). There are four different arrestins–arrestin1 (visual), arrestin2 (β-arrestin1), arrestin3 (β-arrestin2), and arrestin4 (Wilden et al, 1986; Gurevich & Gurevich, 2006; Moore et al, 2007). Arrestin1 and arrestin4 are both visual arrestins and are solely expressed in the retina (Shinohara et al, 1987; Yamaki et al, 1987). β-arrestin1 and β-arrestin2 are ubiquitously expressed, especially in the brain, and play a role in a wide range of cellular processes (Lohse et al, 1990; Attramadal et al, 1992). β-arrestin1 and β-arrestin2 share 78% protein sequence homology and have multiple overlapping roles in various pathways (Attramadal et al, 1992).

Previously, Liu et al (2013) and Thathiah et al (2013) have shown that both β-arrestin1 (Liu et al, 2013) and β-arrestin2 (Thathiah et al, 2013) are increased in AD brains and promotes Aβ production by interacting with the γ-secretase subunit Aph-1 (Liu et al, 2013; Thathiah et al, 2013), thereby linking β-arrestin1 and β-arrestin2 to Aβ pathogenesis. The microtubule-associated protein tau (*MAPT*) plays an essential role in numerous neurodegenerative diseases (Goedert et al, 1988; von Bergen et al, 2001; Lashley et al, 2015), and pathogenic species of tau form neurotoxic aggregates, which correlate with cognitive deficits and neurodegeneration in humans and animal models of tauopathy (Patterson et al, 2011; Ward et al, 2012; Wang & Mandelkow, 2016). Hence, reducing pathogenic tau represents an attractive therapeutic strategy. We have recently shown that β-arrestin2 is also increased in frontotemporal lobar degeneration (FTLD-tau) patients (Woo et al, 2020), and genetic reduction in β-arrestin2 or expression of dominant-negative

[1]Department of Pathology, Case Western Reserve University, School of Medicine, Cleveland, OH, USA   [2]Department of Molecular Medicine, University of South Florida, College of Medicine, Tampa, FL, USA   [3]Department of Molecular Pharmacology and Physiology, University of South Florida, College of Medicine, Tampa, FL, USA   [4]Louis Stokes Cleveland VA Medical Center, Cleveland, OH, USA

Correspondence: jaw330@case.edu

mutants that decreases β-arrestin2 oligomer formation, significantly mitigates tauopathy in vivo (Woo et al, 2020). However, it is not known whether β-arrestin1 could also regulate the pathogenesis of tau. Though β-arrestin1 and β-arrestin2 are functionally similar, they have important distinctions that could result in a lack of or different mechanism of action in pathogenesis (see the Discussion section). It is also unclear what roles β-arrestin1 or β-arrestin2 plays in the GPCR-induced effects on tau. In this study, we found that β-arrestin1 is significantly increased in frontotemporal lobar degeneration-tau (FTLD-tau) patients, a degenerative condition defined by tauopathy in the absence of Aβ deposits, and elevated β-arrestin1 promotes tau accumulation and tauopathy in vitro and in vivo by two distinct mechanisms. Furthermore, we confirmed that both β-arrestin1 and β-arrestin2 mediate GPCR stimulation effects on tauopathy. Therefore, reducing β-arrestin1 or β-arrestin2 is sufficient to block the effects of GPCR stimulation on tauopathy. Here, we further define the molecular mechanistic basis of β-arrestin1 in tauopathy by demonstrating that β-arrestin1 not only induces the dissociation of tau from microtubules but also inhibits tau-induced microtubule assembly. Moreover, we found that β-arrestin1 and β-arrestin2 share a common mechanism to promote aggregation of pathogenic tau by blocking autophagy cargo receptor p62. Indeed, genetic reduction in β-arrestin1 markedly restores synaptic dysfunction and significantly alleviates tauopathy in PS19 transgenic mice in vivo.

# Results

## β2AR and mGluR2 agonists increase pathogenic tau via β-arrestin1 and β-arrestin2

To date, activation of two GPCRs have been implicated in tau phosphorylation and accumulation: β2AR (Dobarro et al, 2013; Luong & Nguyen, 2013; Wisely et al, 2014) and mGluR2 (Lee et al, 2004, 2009). The genetic reduction of β2AR mitigates tauopathy in vivo (Wisely et al, 2014), and β2AR agonist significantly increases amyloid plaques in APPswe/PS1ΔE9 mice (Ni et al, 2006). Furthermore, propranolol, a non-selective β-adrenergic receptor antagonist has been shown to attenuate cognitive impairments in Tg2576 mice (Dobarro et al, 2013). mGluR2 stimulation also significantly increases tau phosphorylation (Lee et al, 2009), and mGluR2/3 antagonist LY-341,495 increases interstitial fluid (ISF) tau (Yamada et al, 2014). Based on these prior observations with animal models and primary neurons, we first set out to confirm agonist-promoted phosphorylation of tau in HeLa cells stably overexpressing tau (V5-tagged 4R0N tau, HeLa-V5-tau cells) and in primary neurons. The β2AR-agonist isoproterenol (ISO) increased phospho-tau in HeLa-V5-tau cells (Fig 1A and B). Therefore, we considered that this effect would be enhanced by transfection of siRNA for β-arrestin1. However, ISO-mediated phosphorylation of tau was ablated by reduction of β-arrestin1, indicating that this signal is mediated by β-arrestin1 as compared to the G protein. In addition to the lack of agonist-promoted tau phosphorylation, baseline tau was reduced by β-arrestin1 reduction

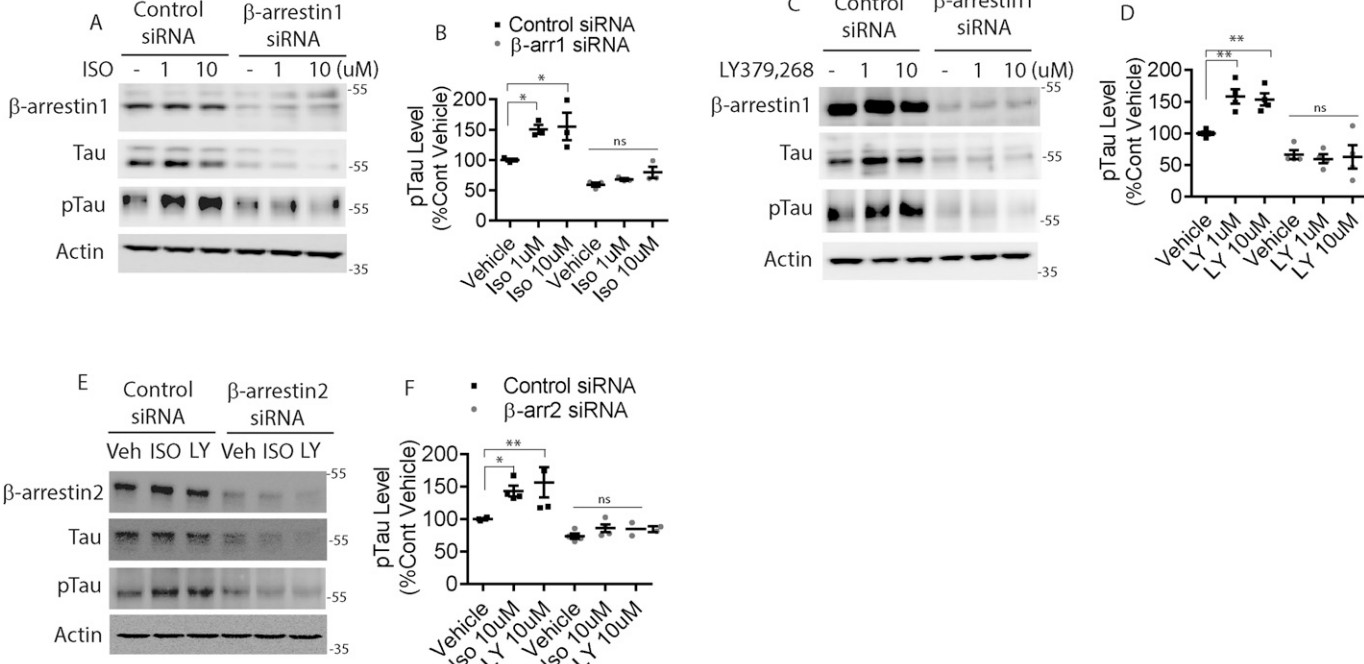

**Figure 1. β-arrestin1 is required for mGluR2- and β2AR-mediated increase in pathogenic tau.**
**(A)** HeLa-V5-tau cells were transfected with control siRNA or β-arrestin1 siRNA. After transfection, the cells were treated with either vehicle or 10 μM isoproterenol (ISO), lysed, and immunoblotted for indicated proteins. Representative blots are shown. **(B)** Quantification of phospho-tau, pS396/pS404-tau (PHF1) levels. n = 3 independent experiments. *P < 0.05. One-way ANOVA with Dunnett's test. **(C)** HeLa-V5-tau cells were transfected with control siRNA or β-arrestin1 siRNA and treated with either vehicle, 1 or 10 μM LY-379,268, lysed, and immunoblotted for indicated proteins. Representative blots are shown. **(D)** Quantification of phospho-tau (PHF1) levels. n = 4 independent experiments. **0 < 0.005. One-way ANOVA with Dunnett's test. **(E)** HeLa-V5-tau cells were transfected with control siRNA or β-arrestin2 siRNA and treated with either vehicle, 10 μM isoproterenol (ISO), or 10 μM LY-379,268 (LY), lysed, and immunoblotted for indicated proteins. **(F)** Quantification of phospho-tau (PHF1) levels. n = 4 independent experiments. *P < 0.05, **P < 0.005. One-way ANOVA with Dunnett's test.

(Fig 1A and B). These same three findings with β-arrestin1 siRNA were also observed in primary neurons of the PS19 transgenic mice: an increase in agonist-promoted phospho-tau in control cells (Fig S1A and B), a loss of this signal with β-arrestin1 reduction, and a decrease in baseline tau (Fig S1E and F). This phenotype was also observed when cells were treated with the mGluR2 agonist LY-379,268 in HeLa-V5-tau cells and the primary neurons (Figs 1C and D and S1C–F). We have not previously studied β-arrestin2 in the context of agonist-promoted tau phosphorylation, and experiments performed with β-arrestin2 siRNA in parallel gave the same three-part phenotype observed with β-arrestin1 reduction (Figs 1E and F and S1G and H) for agonist activation of β2AR or mGluR2. Collectively, these data indicate that the mGluR2 and β2AR-promoted increase in pathogenic tau is dependent on both β-arrestin1 and β-arrestin2, and that these arrestins can also modulate basal levels of tau.

### Elevated β-arrestin1 and colocalization with pathogenic tau (AT8) in FTLD-tau

The finding that β-arrestin1 mediates the increase in pathogenic tau in response to GPCR stimulation prompted us to assess β-arrestin1 levels in FTLD-tau patients. Previously, we have shown that β-arrestin2 levels in the frontal cortex of FTLD-tau patients were significantly increased compare to control subjects (Woo et al, 2020). Compared to control subjects (n = 12), FTLD-tau brains (n = 10) showed >50% increase in β-arrestin1 protein in RIPA-soluble extracts (Fig 2A and B) and RIPA-insoluble extracts (Fig 2C and D). We noted that the levels of insoluble β-arrestin1 mirrored those of insoluble tau in the FTLD-tau brains, with a coefficient of determination ($R^2$ = 0.4874) by linear regression analysis (Fig 2E and F), suggesting a functional association between β-arrestin1 and tau in the tauopathic brain. To assess the spatial relationship between β-arrestin1 and tau, we next stained FTLD-tau frontal gyrus for phospho-tau (AT8 antibody: pS202/pT205-tau) and β-arrestin1. Confocal images of AT8+ tau aggregates, and β-arrestin1, are significantly colocalized (Fig 2G and H) as confirmed by Z-stacked images taken at 1-micron increments (Fig 2G and H). Importantly, we showed the absence of AT8+ tau pathology in control brains, despite the expected detection of β-arrestin1 staining in the same sections (Fig S2A). We also confirmed that secondary antibody only staining failed to show immunoreactivity (Fig S2B). Interestingly, we found that β-arrestin1 mRNA levels are not altered in FTLD-tau frontal gyrus compared to control patients (Fig S2C).

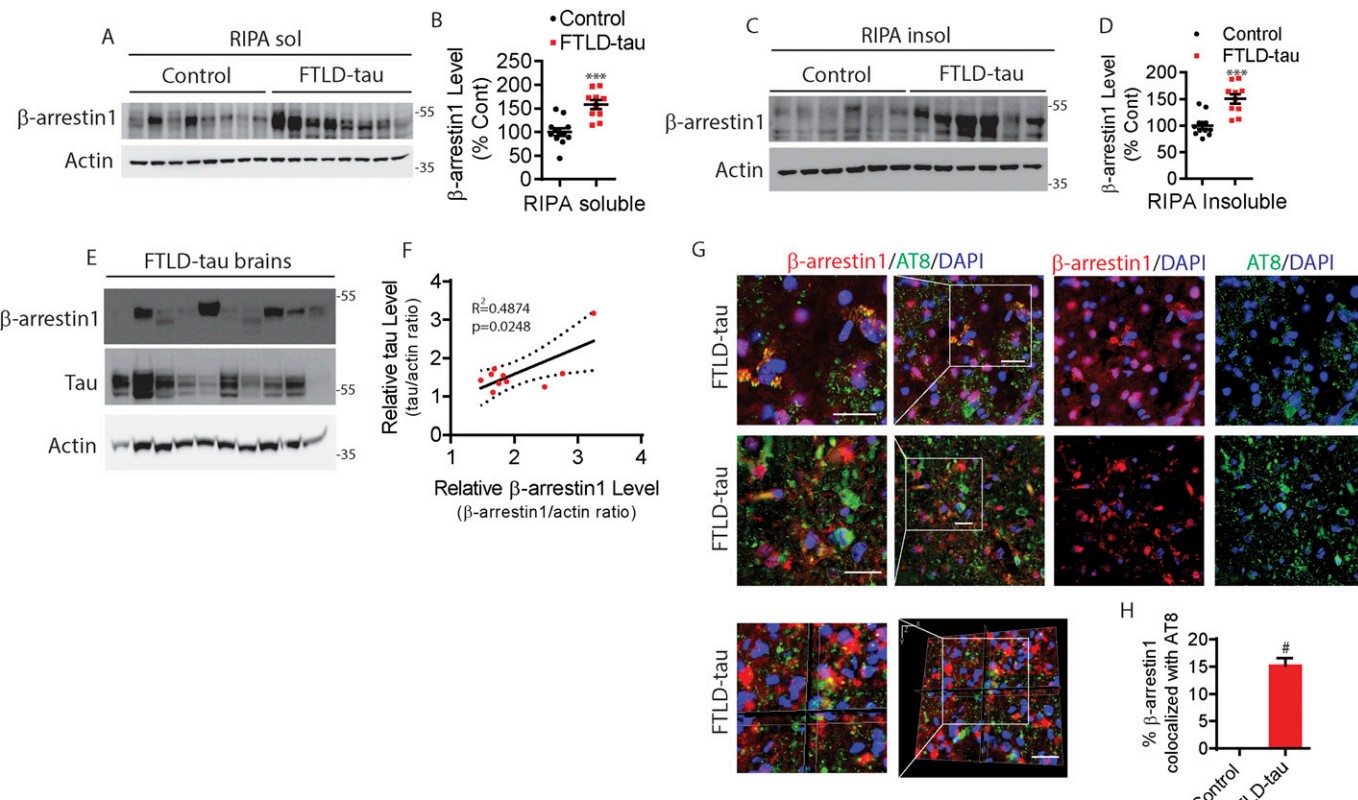

**Figure 2. Elevated β-arrestin1 in FTLD-tau patient brains.**
**(A)** RIPA-soluble extracts from the frontal cortex of healthy controls and FTLD-tau patients were immunoblotted for β-arrestin1 and actin. Representative blots are shown. **(B)** Quantification of RIPA-soluble β-arrestin1 levels. Healthy control (n = 12), FTLD-tau patients (n = 10). ***P < 0.0005. Unpaired t test. **(C)** RIPA-insoluble extracts from the frontal cortex of healthy controls and FTLD-tau patients were immunoblotted for β-arrestin1 and actin. Representative blots are shown. **(D)** Quantification of RIPA-insoluble β-arrestin1 levels. Healthy control (n = 12), FTLD-tau patients (n = 10). ***P < 0.0005. Unpaired t test. **(E)** RIPA-insoluble extracts from the frontal cortex of FTLD-tau patients were immunoblotted for β-arrestin1, tau and actin. **(F)** Correlation between RIPA-insoluble tau and β-arrestin1 in FTLD-tau patients (n = 10 FTLD-tau; R2 = 0.4874, P = 0.0248, linear regression). **(G)** Representative images and Z-stack images of FTLD-tau brains showing that β-arrestin1 and AT8+ (pS202/pT205) tau pathology are colocalized (Scale bar = 20 μm). White boxes are magnified left. **(H)** Quantification of colocalization between β-arrestin1 and AT8. #P < 0.0001. Unpaired t test.

## β-arrestin1 directly promotes the accumulation of pathogenic tau in primary neurons

We also examined the potential for bidirectional control of tau by either knocking down or overexpressing β-arrestin1. Given that β-arrestin1 was significantly increased in FTLD-tau patients, we next assessed whether endogenous β-arrestin1 increases tau levels. Therefore, we transfected HeLa-V5-tau cells with either control siRNA or β-arrestin1 siRNA. As shown by immunoblotting, β-arrestin1 depletion significantly decreased tau (Fig S3A and B). To confirm the relevance of these results in neurons, we used lentivirus-mediated shRNA knockdown of β-arrestin1 in PS19 hippocampal primary neurons. β-arrestin1-shRNA transduced PS19 neurons showed a significant ~50% decrease in immunoreactivity for tau in neuronal cell bodies and processes compared to control shRNA transduced neurons (Fig 3A and B). We confirmed that β-arrestin1-shRNA transduced PS19 cortical primary neurons also exhibited significantly reduced tau levels by Western blotting (Figs 3C and D and S3C). Transient transfection of β-arrestin1 in HeLa-V5-tau cells significantly increased total tau by ~50% (Fig 3E and F) and phospho-tau by nearly twofold (Fig 3E and G). These results collectively show that β-arrestin1 is not only increased and colocalized with pathogenic tau in FTLD-tau brains, but that β-arrestin1–mediated tau regulation underlies both steady-state tau/phospho-tau and GPCR (β2AR & mGluR2) mediated, agonist-promoted, increases in phospho-tau.

## Genetic reduction of *ARRB1* ameliorates tauopathy and cognitive impairments in vivo

We next assessed the physiological effects of reducing endogenous β-arrestin1 on tauopathy in vivo. We crossed PS19 transgenic mice with $Arrb1^{+/-}$ ($ARRB1^{+/-}$) mice to generate PS19 and PS19/$Arrb1^{+/-}$ mice. PS19 mice show tauopathy starting at 4 mo of age, which progressively worsens (Yoshiyama et al, 2007). We first performed immunohistochemistry to detect phospho-tau (pS199/202) from hippocampus of 7-mo-old PS19 and PS19/$Arrb1^{+/-}$ littermates. PS19/$Arrb1^{+/-}$ mice exhibited ~60% reduction in phospho-tau immunoreactivity compared with PS19 littermates (Fig 4A and B). This important finding was further confirmed using sarkosyl extraction of mouse brains. Consistent with the immunohistochemical results, sarkosyl-insoluble tau was significantly reduced by ~40% in PS19/$Arrb1^{+/-}$ compared to PS19 littermates (Fig 4C and E). Sarkosyl-soluble tau was also significantly reduced by ~40% in PS19/$Arrb1^{+/-}$ compared with PS19 littermates (Fig 4C and D). Next, we assessed whether genetic reduction in *ARRB1* rescues impaired spatial learning and memory in PS19 mice using Morris water maze (MWM). Previous studies have shown that PS19 mice exhibit hippocampal-dependent spatial memory deficits around 6 mo of age (Xu et al, 2014; Chalermpalanupap et al, 2018). Therefore, we performed MWM on 6-mo-old WT, PS19, and PS19/$Arrb1^{-/-}$ littermates from PS19/$Arrb1^{+/-}$ crosses with $Arrb1^{+/-}$ mice. Indeed, PS19 mice showed a significantly impaired learning curve compared

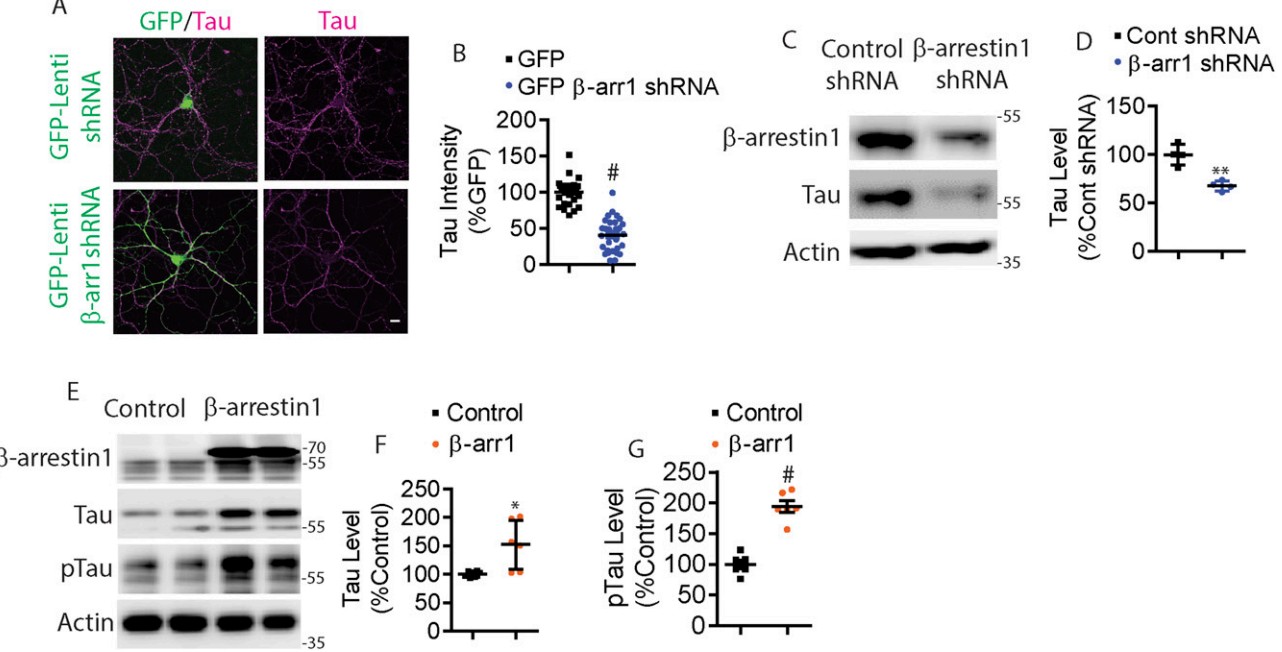

**Figure 3.   β-arrestin1 promotes tau levels.**
**(A)** DIV5 hippocampal primary neurons derived from P0 PS19 mice were transduced with GFP or β-arrestin1-shRNA-GFP lentivirus and immunostained for tau on DIV21. Representative images are shown (Scale bar = 10 μm). **(B)** Quantification of tau intensity. n = 4 independent experiments. #*P* < 0.0001. Unpaired *t* test. **(C)** DIV5 cortical primary neurons derived from P0 PS19 pups were transduced with control or β-arrestin1-shRNA lentivirus. Cortical neurons were lysed at DIV18 and immunoblotted for β-arrestin1, tau, and actin. Representative blots are shown. **(D)** Quantification of total tau levels. n = 4 independent experiments. **P* < 0.005. Unpaired *t* test. **(E)** HeLa-V5-tau cells were transfected with control vector or β-arrestin1, lysed and immunoblotted for β-arrestin1, tau, phospho-tau (AT8), and actin. Representative blots are shown. **(F, G)** Quantification of total tau and phospho-tau (AT8) levels. n = 6 independent experiments. **P* < 0.05, #*P* < 0.0001. Unpaired *t* test.
Source data are available for this figure.

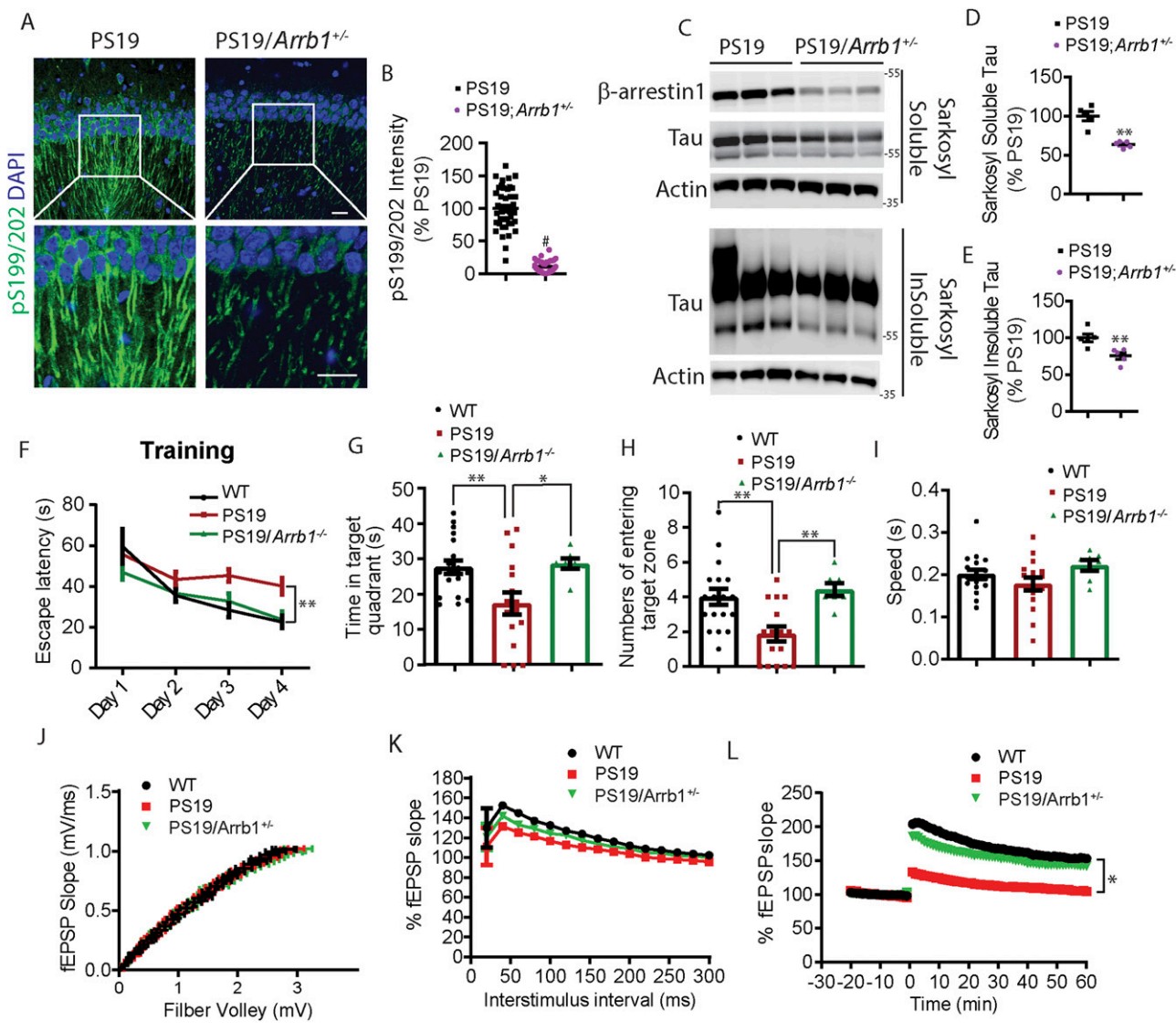

**Figure 4. Genetic reduction in Arrb1 alleviates tauopathy in vivo.**
**(A)** Confocal images of 7-mo-old PS19 and PS19/Arrb1$^{+/-}$ littermates hippocampus stained for phospho-tau, pS199/pS202 (scale bar = 10 $\mu m$). White boxes are magnified below. Representative images are shown. **(B)** Quantification of pS199/pS202 tau intensity. n = 4/genotype. #P < 0.0001. Unpaired $t$ test. **(C)** Sarkosyl-soluble and insoluble extracts from 7-mo-old PS19 and PS19/Arrb1$^{+/-}$ brains were immunoblotted for β-arrestin1, tau, and actin. Representative blots are shown. **(D, E)** Quantification of sarkosyl-soluble and insoluble tau levels. n = 5/genotype. **P < 0.005. Unpaired $t$ test. **(F)** Morris water maze shows the latency of 6-mo-old PS19 and PS19/Arrb1$^{-/-}$ littermates during 4-d training test (WT:18 mice, PS19:16 mice, and PS19/Arrb1$^{-/-}$:7 mice). **P < 0.005. Two-way ANOVA with Dunnett's post hoc test. **(G)** Quantification of time spent in target quadrant (s) during 24-h probe. **P < 0.005,*P < 0.05. One-way ANOVA with Dunnett's post hoc test. **(H)** Quantification of number of times entered into the target zone during 24-h probe. **P < 0.005. One-way ANOVA with Dunnett's post hoc test. **(I)** Quantification of average swim speed (s). No significant difference between all genotypes. **(J)** Input–output analysis from 4-mo-old WT, PS19, and PS19/Arrb1$^{+/-}$ acute slices. (WT: 33 slices, 5 mice; PS19: 32 slices, 5 mice; PS19/Arrb1$^{+/-}$: 30 slices, 4 mice). No significant differences observed. **(K)** Paired pulse facilitation analysis from 4-mo-old WT, PS19, and PS19/Arrb1$^{+/-}$ acute slices. (WT: 33 slices, 5 mice; PS19: 40 slices, 5 mice; PS19;Arrb1$^{+/-}$:35 slices, 4 mice). Two-way ANOVA with Dunnett's post hoc test. #P < 0.0001: 20–180 ms, WT versus PS19. ***P < 0.0005: 20 ms, PS19 versus PS19/Arrb1$^{+/-}$. **P < 0.005: 200–260 ms, WT versus PS19. *P < 0.05: 280–300 ms, WT versus PS19; 40,100, & 120 ms, PS19 versus PS19/Arrb1$^{+/-}$. **(L)** Long-term potentiation induced by theta burst stimulation. (WT: 31 slices, 5 mice; PS19: 29 slices, 5 mice; PS19/Arrb1$^{+/-}$:31 slices, 4 mice). Two-way ANOVA with Dunnett's post hoc test. *P< 0.0001 in all time points, WT versus PS19 and PS19/Arrb1$^{+/-}$ versus PS19. All data are presented as mean ± SEM.

with WT littermates (Fig 4F). However, PS19/Arrb1$^{-/-}$ littermates showed an indistinguishable learning curve compared with WT littermates (Fig 4F). Furthermore, we found that the target quadrant occupancy of PS19/Arrb1$^{-/-}$ mice was significantly higher than PS19 littermates in the probe trial (Fig 4G and H),

indicating that genetic reduction in Arrb1 rescues impaired spatial memory. We further confirmed that there were no genotype-dependent differences in average swimming speed (Fig 4I) to show that differences in latency were not due to differences in locomotor activity.

## Genetic reduction of *ARRB1* rescues functional synaptic deficits in PS19 mice

The initial characterization of the PS19 mice demonstrated impaired paired-pulse facilitation (PPF) and long-term potentiation (LTP) at 6 mo of age (Yoshiyama et al, 2007). Our studies later showed that PS19 mice exhibit pronounced LTP defects as early as 3 mo of age (Woo et al, 2017b, 2019). To assess functional changes in synaptic plasticity, we performed electrophysiological recordings of the CA3-CA1 Schaffer collateral pathway of acute brain slices of 4-mo-old wild-type, PS19, and PS19/*Arrb1*$^{+/-}$ mice. Initial input–output (IO) analysis indicated no significant differences among WT, PS19, and PS19/*Arrb1*$^{+/-}$ littermate slices (Fig 4J). In PPF experiments, we observed significant reductions in fEPSP slope in PS19 slices in all interstimulus intervals compared with wild-type slices, which was accentuated in earlier interstimulus intervals (Fig 4K) similar to that previously reported (Woo et al., 2017b, 2019). In PS19/*Arrb1*$^{+/-}$ slices, we observed significantly stronger PPF at interstimulus intervals ranging from 20 to 120 ms compared with PS19, indicating a partial rescue. LTP recordings using theta-burst stimulation showed PS19 slices to be strongly impaired in both induction and maintenance of LTP compared with wild-type slices (Fig 4L). However, PS19/*Arrb1*$^{+/-}$ slices showed significantly restored LTP compared with PS19 littermates, nearly to the level of wild-type slices (Fig 4L). These functional synaptic plasticity results were corroborated in mature DIV21 primary hippocampal neurons stained for synaptophysin. Specifically, PS19 hippocampal primary neurons (control GFP transduced) exhibited significantly reduced synaptophysin immunoreactivity in primary neurites compared with that in wild-type neurites (control GFP transduced). In contrast, PS19 neurons transduced with *β*-arrestin1-shRNA-GFP significantly restored synaptophysin immunoreactivity (Fig S4A and B).

## *β*-arrestin1 promotes the dissociation of tau from microtubules and inhibits tau-induced microtubule assembly

Tau is a microtubule-associated protein that stabilizes microtubules (Cleveland et al, 1977). However, in tauopathies like AD, tau first dissociates from microtubules, mislocalizes from somatoaxonal to somatodendritic compartments (Biernat & Mandelkow, 1999; Ballatore et al, 2007), and becomes progressively insoluble to ultimately form filamentous aggregates (Alonso et al, 1997). Because *β*-arrestin1 binds directly to microtubules and recruits Mdm2 and ERK2 (Hanson et al, 2007a; Gurevich & Gurevich, 2014), we first assessed whether the *β*-arrestin1 and tau "compete" for binding to microtubules. 1 *μ*g of recombinant His-tau (4R) was incubated with purified microtubules plus BSA (control) or recombinant purified *β*-arrestin1 for 30 min. After incubation, the sample was subjected to centrifugation at 100,000*g*. Here, the supernatant contains microtubule-unbound proteins and the pellet contains microtubule-bound proteins. Remarkably, *β*-arrestin1 significantly reduced the amount of tau bound to microtubules by ~45%, while increasing the amount of free tau in the supernatant (Fig 5A and B). The inhibitory effect of *β*-arrestin1 on tau binding to microtubules was dose-dependent, as increasing amounts of *β*-arrestin1 progressively reduced tau bound to microtubules (Fig 5C). Next, we assessed whether *β*-arrestin1 alters tau-induced microtubule assembly in vitro. As expected, tubulin alone exhibited time-dependent

polymerization into microtubules, which greatly accelerated with the addition of recombinant tau (Fig 5D). However, including *β*-arrestin1 together with tau fully inhibited tau-induced acceleration of microtubule assembly (Fig 5D). Addition of *β*-arrestin1 alone with tubulin also weakly reduced tubulin polymerization compared with tubulin alone, suggesting that *β*-arrestin1/tubulin binding may have a minor inhibitory role in microtubule assembly.

To determine whether the inhibitory action of *β*-arrestin1 in tau-dependent microtubule assembly observed in vitro occurs in cells, we transfected HeLa-V5-tau cells with control siRNA or *β*-arrestin1 siRNA. After transfection, cells were treated with nocodazole for 30 min, which induces the rapid disassembly of microtubules (Woo et al, 2019). After 30 min, we washed out the media containing nocodazole and allowed cells to recover for 1 h. Upon nocodazole treatment, staining for tubulin appeared highly disorganized in control or *β*-arrestin1 siRNA transfected conditions (Fig 5E). Upon washout of nocodazole for 1 h, reassembly of microtubules was readily visible as seen by salient filamentous microtubule staining in perinuclear regions, which was significantly increased by nearly threefold in *β*-arrestin1 siRNA transfected cells compared to control siRNA transfected cells (Fig 5E and F). We also found that the colocalization between tau and MAP2 are reduced in *β*-arrestin1-shRNA transduced PS19 primary neurons compared with control shRNA transduced neurons (Fig S5). These in vitro and in cellulo results indicate that *β*-arrestin1 promotes the dissociation of tau from microtubules, which both inhibits microtubule assembly and enables tau missorting.

## *β*-arrestin1 increases tau by disrupting p62 self-interaction and impeding p62 flux

Although tau dissociation from microtubules by *β*-arrestin1 appears to deregulate microtubule dynamics leading to tau missorting and aggregation, this mechanism nevertheless does not readily explain the increase in total tau due to increased *β*-arrestin1. No changes in tau mRNA were observed either after *β*-arrestin1 overexpression or knockdown (Fig 6A and B). We next assessed whether endogenous *β*-arrestin1 alters tau turnover. Indeed, cycloheximide (CHX) chase experiments showed that *β*-arrestin1 siRNA significantly facilitates the turnover of tau (Fig 6C and D), indicating that endogenous *β*-arrestin1 enhances tau levels by increasing its stability.

Multiple studies have shown that microtubule destabilization impairs autophagosome maturation and autophagy-mediated protein degradation (Aplin et al, 1992; Fass et al, 2006), as microtubule-based transport is needed for the delivery of autophagosomes to lysosomes (Boecker & Holzbaur, 2019; Farfel-Becker et al, 2019). To clear misfolded proteins through autophagy, autophagy cargo receptors such as p62/SQSTM1 must first sequester the cargo and link the polyubiquitinated cargo to LC3+ autophagosomes, after which they are collectively delivered to lysosomes for fusion and degradation (Pankiv et al, 2007; Katsuragi et al, 2015). As *β*-arrestin1 displaced tau from microtubules, destabilized microtubules, and also increased tau stability, we initially hypothesized that *β*-arrestin1–mediated destabilization of microtubules could disrupt the delivery of p62 to LC3+ autophagosomes, thereby increasing tau stability. HeLa-V5-tau cells were

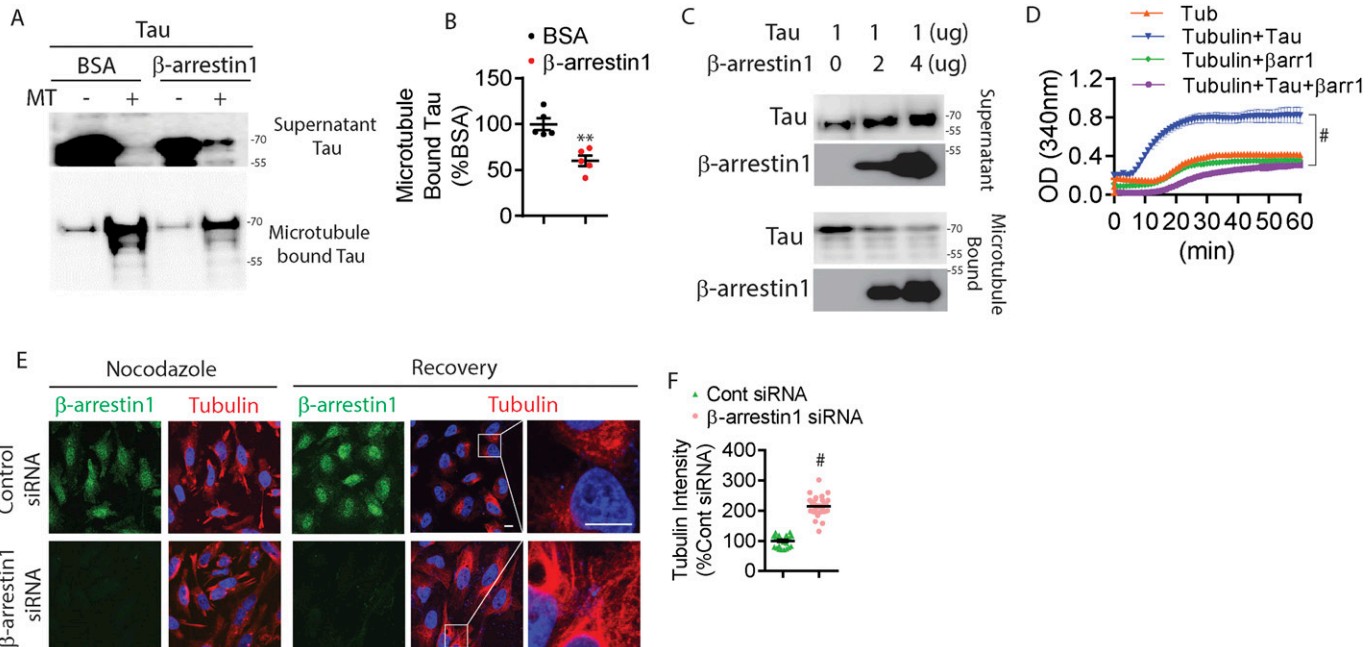

**Figure 5. β-arrestin1 inhibits tau-induced tubulin polymerization and disrupts microtubule stability.**
**(A)** Microtubule-binding sedimentation assay was performed using 1 μg recombinant tau incubated with either 2 μg BSA or 2 μg recombinant β-arrestin1. Indicated amounts of β-arrestin1 or BSA were incubated with or without 0.4 nM pre-polymerized microtubules in the presence of tau, and microtubule-bound tau was monitored by co-sedimentation and subsequent immunoblotting for tau. Representative blots are shown. **(B)** Quantification of microtubule-bound tau. n = 5 independent experiments. **P < 0.005. Unpaired t test. **(C)** Microtubule-binding sedimentation assay with indicated amount of recombinant β-arrestin1, 1 μg of tau, and 0.4 nM pre-polymerized microtubules. Tau and β-arrestin1 in the microtubule-bound pellet and unbound supernatant were examined by co-sedimentation and subsequent immunoblotting. **(D)** Tubulin polymerization was measured by turbidity at 340 nm in the presence of 2 μg of indicated recombinant proteins. n = 4 independent experiments. #P < 0.0001. Two-way repeated-measures ANOVA. **(E)** Confocal images of HeLa-V5-tau cells transfected with either control siRNA or β-arrestin1 siRNA and treated with 20 μM of nocodazole for 30 min and recovered for another 1 h. Cells were fixed and immunostained for tubulin and β-arrestin1 (Scale bar = 10 μm). White boxes magnified to the right. **(F)** Quantification of transfected cells with tubulin intensity normalized to control siRNA transfected cells. n = 4 independent experiments. #P < 0.0001. Unpaired t test.

transfected with GFP-p62 together with control vector or β-arrestin1 and cells were treated with vehicle or nocodazole for 30 min. In control vector-transfected cells, GFP-p62 puncta (green) of varying sizes were present, whereas endogenous LC3 (magenta) was observed as small punctate staining (Fig 6E). GFP-p62 often colocalized (white puncta) with LC3 (Fig 6E). As expected, nocodazole treatment decreased GFP-p62 colocalization with endogenous LC3-positive puncta in vector control transfected cells (Fig 6E and F). β-arrestin1–overexpressing HeLa-V5-tau cells showed marked disruption of GFP-p62 colocalization with LC3 at steady state, to an extent that was equivalent to that observed with nocodazole treatment (Fig 6E and F). Hence, nocodazole treatment to β-arrestin1–expressing HeLa-V5-tau cells did not further disrupt GFP-p62/LC3 colocalization (Fig 6E and F). These data indicated that either β-arrestin1–mediated destabilization of microtubules is as severe as nocodazole treatment (unlikely) or other factors may be at play in such robust disruption of p62-LC3 colocalization. To examine LC3 and p62 in a different way, we assessed the effects of β-arrestin1 on LC3 and GFP-p62 puncta with or without bafilomycin A1 treatment, a potent lysosome inhibitor known to promote the accumulation of both LC3 and p62 puncta (Yoshimori et al, 1991; Yamamoto et al, 1998; Mauvezin & Neufeld, 2015). Overexpression of β-arrestin1 in HeLa-V5-tau cells not only reduced LC3 puncta at steady state but also significantly blunted bafilomycin A1–induced increase in LC3 puncta (Fig S6A and B), indicating that β-arrestin1

blocks autophagy at the level of LC3 or upstream. Likewise, over-expression of β-arrestin1 also reduced GFP-p62 puncta at steady-state and significantly blunted bafilomycin A1–induced increase in GFP-p62 puncta (Fig 6G and H). Moreover, whereas bafilomycin A treatment significantly increased the colocalization of GFP-p62 with LC3 in vector control transfected cells, β-arrestin1 transfection significantly blunted the increase in GFP-p62/LC3 colocalization (Fig 6G and I). Taken together, these results indicate that β-arrestin1 blocks autophagy at the level of p62 or upstream and likely not directly on LC3.

P62 is associated with neurofibrillary tangles (Kuusisto et al, 2002; King et al, 2013), and soluble cytoplasmic p62 levels are significantly reduced in AD brains (Zheng et al, 2012). Increased p62 expression improves cognitive impairments in AD animal models by enhancing autophagy induction (Babu et al, 2008; Zheng et al, 2012). To further investigate β-arrestin1–induced changes in p62, we assessed p62 flux using the mCherry-GFP-p62 reporter. This reporter takes advantage of the sensitivity of GFP (green) and the insensitivity of mCherry (pseudocolored to magenta) to low pH, which allows the tracking of p62 flux to lysosomes (Pankiv et al, 2007; Larsen et al, 2010). Thus, colocalized mCherry and GFP (white or light green) are indicative of non-lysosomal LC3. However, upon fusion with acidified lysosomes (autolysosomes), mCherry puncta persist, whereas GFP puncta disappear (hence magenta only). We co-transfected HeLa-V5 cells with mCherry-GFP-p62 with either

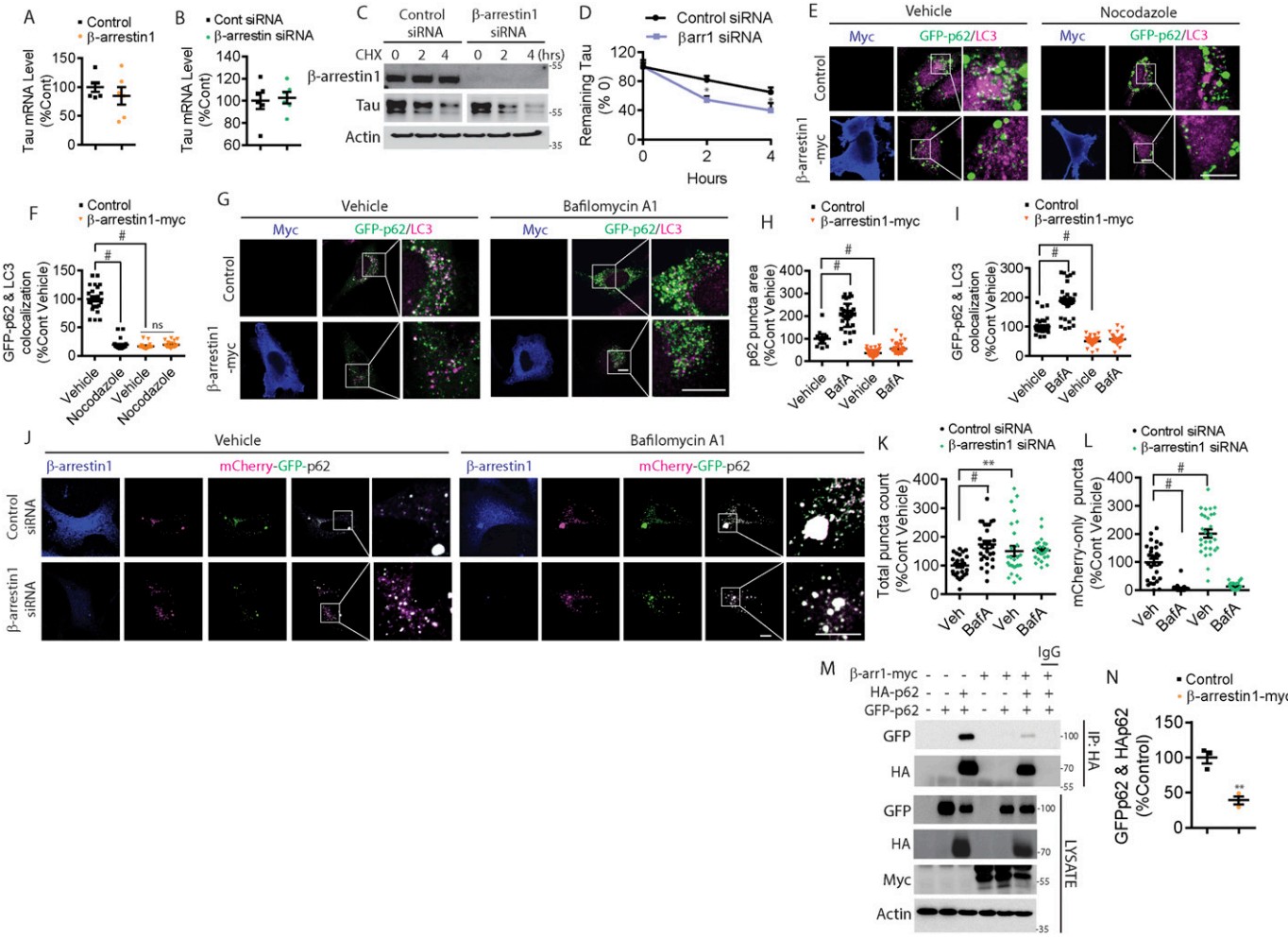

**Figure 6. β-arrestin1 inhibits autophagy-mediated tau clearance.**
**(A)** Quantification of β-arrestin1 mRNA levels by qRT-PCR in HeLa-V5-tau cells transfected with control vector or β-arrestin1. n = 6 independent experiments.
**(B)** Quantification of β-arrestin1 mRNA levels by qRT-PCR in HeLa-V5-tau cells transfected with either control siRNA or β-arrestin1 siRNA. n = 6 independent experiments.
**(C)** HeLa-V5-tau cells were transfected with control siRNA or β-arrestin1 siRNA and treated with cycloheximide (100 µg/ml) for 2 and 4 h. Cells were subjected to immunoblotting for β-arrestin1, tau, and actin. Representative blots are shown. **(D)** Quantification of tau remaining after cycloheximide treatment. n = 3 independent experiments. *P < 0.05. Two-way repeated-measures ANOVA. **(E)** Confocal images of HeLa-V5-tau cells transfected with GFP-p62 with either control vector or β-arrestin 1-myc and treated with vehicle or 20 µM of nocodazole for 30 min. Cells were fixed and immunostained for myc and LC3 (Scale bar = 10 µm). Representative images are shown. **(F)** Quantification of GFP-p62 and LC3 colocalization. n = 4 independent experiments. #P < 0.0001. One-way ANOVA with Dunnett's post hoc test. **(G)** Confocal images of HeLa-V5-tau cells transfected with GFP-p62 and either vector control or β-arrestin1-myc and treated with vehicle or 100 nM of Bafilomycin A1 for 4 h. Cells were fixed and immunostained for myc and LC3 (Scale bar = 10 µm). Representative images are shown. **(H)** Quantification of GFP-p62 puncta area. n = 4 independent experiments. #P < 0.0001. One-way ANOVA with Dunnett's post hoc test. **(I)** Quantification of GFP-p62 and LC3 colocalization. n = 4 independent experiments. #P < 0.0001. One-way ANOVA with Dunnett's post hoc test. **(J)** Confocal images of HeLa-V5-tau cells transfected with mCherry-GFP-p62 with either control siRNA or β-arrestin1 siRNA and treated with vehicle or 100 nM of Bafilomycin A1 for 4 h. Cells were fixed and immunostained for β-arrestin1 (scale bar = 10 µm). mCherry is pseudocolored to magenta. Representative images are shown. **(K)** Quantification of total p62 puncta (mCherry+GFP) normalized to control vehicle treatment. n = 4 independent experiments. #P < 0.0001, **P < 0.005. One-way ANOVA with Dunnett's test. **(L)** Quantification of mCherry-only (magenta) puncta normalized to control vehicle treatment. n = 4 independent experiments. #P < 0.0001. One-way ANOVA with Dunnett's test. **(M)** HeLa-V5-tau cells were transiently transfected with control vector or β-arrestin1-myc together with either GFP-p62 and/or HA-p62 and subjected to co-immunoprecipitation for HA and immunoblotting for GFP, HA, myc, and actin. Representative blots are shown. **(N)** Quantification of GFP-p62 and HA-p62 interaction n = 3 independent experiments. **P < 0.005. Unpaired t test.

control siRNA or β-arrestin1 siRNA and quantified total mCherry+GFP puncta (white/light green) and mCherry-only (magenta) puncta. As expected, bafilomycin A treatment increased total mCherry+GFP puncta in vector control transfected cells (Fig 6J and K). However, β-arrestin1 siRNA transfected cells showed significantly increased total mCherry+GFP puncta at steady state, and bafilomycin A treatment did not significantly further increase this measure

(Fig 6J and K). The percentage of acidified mCherry-only (magenta) puncta out of total p62 puncta was increased by ~2-fold in β-arrestin1 siRNA versus control siRNA transfected cells, nearly all of which were abolished by 4 h bafilomycin A treatment, indicating that the loss of β-arrestin1 promotes p62 flux (Fig 6J and L). β-arrestin1 formed a specific complex with HA-p62 in co-immunoprecipitation (co-IP) experiments from HeLa-V5-tau cells

(Fig S6C), suggesting that β-arrestin1 might directly modify p62 activity by physical interaction. To query this possibility, we took advantage of the known ability of p62 to form particles by self-interaction via its N-terminal PB1 domain, which allows the formation of p62 helical filaments arranged in a head to tail configuration (Ciuffa et al, 2015), a step that is essential for its cargo receptor activity (Itakura & Mizushima, 2011; Wurzer et al, 2015). Hence, we tested whether β-arrestin1/p62 interaction alters the ability of p62 to form self-interacting complexes using HA-p62 and GFP-p62 constructs. We observed the specific presence of GFP-p62 in HA-p62 immune complexes, which was significantly diminished by >60% by β-arrestin1 overexpression under conditions where similar amounts of HA-p62 were pulled down in HA immune complexes (Fig 6M and N). These results therefore show that increased β-arrestin1, as seen in brains of FTLD-tau and AD, strongly blocks the self-interaction of p62, an initial step required for p62-mediated clearance of cargo including misfolded tau (Babu et al, 2008; Itakura & Mizushima, 2011; Zheng et al, 2012; Wurzer et al, 2015).

## Discussion

Previous studies have implicated multiple GPCR pathways in AD pathogenesis (Lee et al., 2004, 2009; Minkeviciene et al, 2004; Sun et al, 2005; Ni et al, 2006; Bakshi et al, 2008; Thathiah et al, 2009; AbdAlla et al, 2009a, 2009b; Alley et al, 2010; Dobarro et al, 2013; Luong & Nguyen, 2013; Wisely et al, 2014), including β2AR (Kalaria et al, 1989; Dobarro et al, 2013; Luong & Nguyen, 2013; Wisely et al, 2014) and mGluR2 (Lee et al, 2004, 2009). β2AR is significantly increased in the frontal cortex and hippocampus in AD brains compared with controls (Kalaria et al, 1989). Genetic studies have shown that polymorphisms in β2AR are associated with higher risk for developing sporadic AD (Rosenberg et al, 2008; Yu et al, 2008), and genetic reduction in β2AR significantly mitigates tauopathy in vivo (Wisely et al, 2014). Isoproterenol, the classic β2AR agonist, markedly increases tau phosphorylation thereby inducing memory deficits in rats (Sun et al, 2005). mGluR2 is also significantly increased in AD, and mGluR2 expression closely correlates with hyperphosphorylated tau deposition (Lee et al., 2004, 2009). The mGluR2 agonist LY-379,268 has been reported to increase tau phosphorylation via ERK activation (Lee et al, 2009). However, it is unclear how different classes of GPCRs similarly affect AD pathogenesis. In fact, β2ARs couple to Gαs, increasing intracellular cAMP by activating adenylyl cyclase, whereas mGlu2R couples to Gαi, inhibiting adenylyl cyclase and lowering intracellular cAMP. β-arrestin1 and β-arrestin2 were initially identified and named because of their actions to rapidly attenuate GPCR signaling through agonist-promoted, GRK-mediated, receptor uncoupling from the G protein (Wilden et al, 1986; Lohse et al, 1990; Gurevich & Gurevich, 2006; Moore et al, 2007). It is now recognized that they act as scaffolds, adapters, and chaperones, leading to receptor internalization as well as de novo, G protein–independent, signaling (Attramadal et al, 1992; Lefkowitz et al, 2006; Smith & Rajagopal, 2016). We first sought to determine whether β-arrestin1 and/or β-arrestin2 act as a point of convergence by which β2AR and mGluR2 agonism alters tau phosphorylation. If receptor function

was enhanced by reduction of either arrestin, then we expected agonist-mediated tau phosphorylation to be enhanced. Our findings in fact showed that the loss of β-arrestin1 or β-arrestin2 ablates β2AR or mGluR2 agonist–dependent increases in phospho-tau, indicating that these receptors are transducing this signal via these arrestins. Although it remains to be determined whether other AD-implicated GPCRs (i.e., ADRBs, GPR3, AT2R, and CXCR2) (Bakshi et al, 2008; Thathiah et al, 2009; AbdAlla et al, 2009a, 2009b) require β-arrestin1 and/or β-arrestin2 for their pathogenic activities, these initial observations led us to examine brains of FTLD-tau patients for alterations in β-arrestin1 levels. Interestingly, previous studies had reported that β-arrestin1 (Liu et al, 2013) and β-arrestin2 (Thathiah et al, 2013) are significantly increased in brains of AD patients and that both β-arrestin1 and β-arrestin2 interact with the Aph-1 subunit of the γ-secretase complex to increase Aβ production, thereby linking β-arrestin1 and β-arrestin2 to Aβ pathogenesis. We recently showed that β-arrestin2 is significantly elevated in brains of FTLD-tau patients, and increased β-arrestin2 promotes tau aggregation in the absence of GPCR stimulation (Woo et al, 2020).

Here, we report that β-arrestin1 levels are highly elevated in brains of FTLD-tau patients, a disease pathologically defined by tauopathy in the absence of Aβ deposits (Irwin et al, 2015). Moreover, the observation that insoluble tau levels correlate with insoluble β-arrestin1 levels and that AT8-positive phospho-tau aggregates are nearly perfectly colocalized with β-arrestin1, suggests a functional pathogenic relationship between β-arrestin1 and tau pathogenesis in FTLD-tau.

The above findings led us to hypothesize that increased β-arrestin1 levels promote tau accumulation and tauopathy, whereas reduced β-arrestin1 levels counteract such phenotypes in primary neurons and in vivo. This hypothesis was confirmed in primary neurons by β-arrestin1 overexpression and RNAi-mediated silencing experiments. In vivo, genetic reduction of ARRB1 not only alleviated tauopathy in PS19 transgenic mice but also functionally rescued the prominent deficits in synaptic plasticity (i.e., PPF and LTP) and synaptic integrity in PS19 acute slices and neurons. These findings therefore indicate that pathogenic tau accumulation upregulates β-arrestin1 through as yet unknown mechanisms, which in turn, further drives tauopathy. Hence, the observation that a 50% reduction in ARRB1 ameliorates tauopathy and associated synaptic dysfunction demonstrates the proof-of-principle that β-arrestin1 represents a viable point of therapeutic interdiction to break this pathogenic feed-forward loop.

A major biological function of tau is ascribed to its ability to bind and stabilize microtubules as well as promote their assembly (Cleveland et al, 1977). In AD and other tauopathies, however, tau dissociates from microtubules, leading to its missorting from the somatoaxonal to somatodendritic compartments (Biernat & Mandelkow, 1999; Ballatore et al, 2007; Hoover et al, 2010). This event occurs early in the disease process and is thought to be required for its hyperphosphorylation and self-assembly into aggregates (Wang & Mandelkow, 2016). As a significant pool of β-arrestin1 binds directly to microtubules (Hanson et al, 2007a; Gurevich & Gurevich, 2014), we show here for the first time that β-arrestin1 binding to microtubules promotes the dissociation of tau from microtubules in a dose-dependent manner, thereby potently inhibiting tau-mediated

microtubule assembly in vitro and in transfected cells. Such actions of β-arrestin1 are highly reminiscent of the manner which the actin-binding protein cofilin displaces tau from microtubules, inhibits tau-induced microtubule assembly, and promotes tauopathy (Woo et al, 2019). Interestingly, β-arrestin1 and β-arrestin2 bind to cofilin and scaffold the interaction with the cofilin activating phosphatase chronophin to enhance cofilin activation (Zoudilova et al, 2007; Zoudilova et al, 2010). β-arrestin2 interaction with cofilin also plays an important role in the translocation of activated cofilin to dendritic spines to regulate spine morphology (Pontrello et al, 2012). However, β-arrestin1 inhibited tau microtubule binding and tau-induced microtubule assembly in the setting of purified recombinant proteins where cofilin was absent. Hence, such inhibitory actions of β-arrestin1 do not require cofilin per se and support the notion that the capacity of β-arrestin1 to displace tau from microtubules (with or without cofilin) contributes to tau mislocalization and subsequent propensity to self-assemble into aggregates.

Microtubule dynamics and autophagy machinery are intricately linked in cells and particularly in neurons, as autophagosomes formed in distal neurites or axons must come together with mature lysosomes that are relatively enriched in the soma (Lee et al, 2011; Maday et al, 2012; Cheng et al, 2015; Wang et al, 2015). Such spatial disparity therefore necessitates microtubule-based transport of autophagosomes and lysosomes over relatively long distances. Indeed, impaired microtubule dynamics disrupts autophagic clearance (Aplin et al, 1992; Fass et al, 2006; Boecker & Holzbaur, 2019; Farfel-Becker et al, 2019), and defects in autophagy contribute to AD pathogenesis (Nixon et al, 2005; Yang et al, 2008; Sanchez-Varo et al, 2012) by promoting the accumulation of Aβ and tau (Babu et al, 2008; Zheng et al, 2012; Xu et al, 2019). Having observed that β-arrestin1 does not alter tau mRNA but increases tau protein stability, we initially focused on the p62-LC3 autophagy machinery because microtubule-based transport facilitates the coming together of p62-bound cargo with LC3+ autophagosomes (Lee et al, 2011; Maday et al, 2012; Cheng et al, 2015; Wang et al, 2015). Moreover, we recently reported that β-arrestin2 disrupts p62-mediated tau clearance (Woo et al, 2020). Thus, we hypothesized that β-arrestin1 could inhibit p62-mediated tau clearance as β-arrestin1 and β-arrestin2 share multiple biological functions with 78% sequence identity.

p62/SQSTM1 knockout mice display severe neurodegeneration as well as hyperphosphorylated tau and neurofibrillary tangles (Babu et al, 2008), and p62 overexpression strongly reduces pathogenic tau in transfected cells and in vivo (Xu et al, 2019). We found that β-arrestin1 overexpression alone is as effective as nocodazole treatment (potent microtubule destabilizing agent) in disrupting p62-LC3 colocalization at steady-state, which suggests that either β-arrestin1 is as effective as nocodazole in destabilizing microtubules (which is unlikely) or other mechanisms might also contribute to such robust disruption. Indeed, our finding that β-arrestin1 reduces both LC3 and p62 puncta and significantly blunts bafilomycin-induced accumulation of both LC3 and p62 puncta indicates that β-arrestin1 inhibits autophagy at the level of p62 per se or upstream. P62 flux and co-IP experiments confirmed that β-arrestin1 acts to inhibit autophagy at the level of p62, as β-arrestin1 not only impedes p62 flux but also binds to p62 and

interferes with p62 self-association, an essential step for the formation of p62 bodies (Ciuffa et al, 2015). Such self-association of p62 via its N-terminal PB1 domain is essential for its cargo receptor activity by enabling stronger connection (multiple binding) to its ubiquitinated cargo as well as simultaneous binding to multiple LC3 proteins (Itakura & Mizushima, 2011; Wurzer et al, 2015), which helps to account for the loss of p62 puncta seen by β-arrestin1 overexpression. Moreover, cargo-bound p62 acts to promote autophagosome formation by enhancing the conversion of LC3 to its active lipidated form LC3-II (Cha-Molstad et al, 2017), which likely accounts for the observation that β-arrestin1 overexpression also decreases LC3 puncta and reduces p62-LC3 colocalization. Such mechanisms of β-arrestin1 in binding to p62 and interfering with p62 self-oligomerization, together with destabilization of microtubules, are consistent with the observed role of β-arrestin1 in impeding p62 flux and impairing the clearance of misfolded tau.

To date, no previous study has implicated β-arrestin1 in tauopathy, microtubule dynamics, or p62-mediated autophagy. Furthermore, it has not been explored whether β-arrestin1 or β-arrestin2 is required for GPCR-induced tau phosphorylation. Our findings collectively implicate β-arrestin1 in several events that promote tauopathy: transducing the agonist-occupied GPCR signal to tau phosphorylation, destabilization of microtubules which releases tau and promotes tau mislocalization, and inhibition of p62-mediated tau clearance (Fig 7A and B).

β-arrestin1 and β-arrestin2 share 78% protein sequence homology, thereby sharing multiple biological functions, including GPCR desensitization and internalization. Double knockout of both genes causes neonatal death in mice (Zhang et al, 2010), whereas single knockout of either β-arrestin1 (Conner et al, 1997) or β-arrestin2 (Bohn et al, 1999) are relatively normal, suggesting that β-arrestin1 and β-arrestin2 play additive roles in their shared biological functions. As such, β-arrestin1 and β-arrestin2 may act in an additive fashion to drive tauopathy. However, it is also recognized that the two β-arrestins have differences in their functional capacities. β-arrestin2 has sixfold greater affinity for clathrin (Goodman et al, 1996) and is 10-fold more efficient at internalizing β2AR (Kohout et al, 2001). A 100-fold greater amount of β-arrestin1 is required to reconstitute wild-type β2AR internalization compared with β-arrestin2 in β-arrestin1/2 double-knockout mouse embryonic fibroblasts (Kohout et al, 2001). In contrast, the angiotensin II type 1A receptor appears to be more sensitive to β-arrestin1 (Kohout et al, 2001). Multiple studies have shown that β-arrestin1 and β-arrestin2 constitutively form homo- and hetero-oligomers (Storez et al, 2005; Milano et al, 2006), and these oligomers are unable to bind to GPCRs (Gurevich & Benovic, 1993; Palczewski et al, 1994; Gurevich et al, 1995; Pulvermuller et al, 2000; Dinculescu et al, 2002; Boularan et al, 2007; Hanson et al, 2007b; Vishnivetskiy et al, 2011; Breitman et al, 2012; Kim et al, 2012; Zhuang et al, 2013). Recently, we showed that β-arrestin2 oligomers mediate tauopathy by impairing pathogenic tau clearance (Woo et al, 2020), and β-arrestin2 oligomerization mutants show dominant-negative "anti-oligomer" effects in vitro and in vivo. However, further studies are necessary to determine whether β-arrestin1 oligomerization mutants also act as "anti-oligomer" dominant-negative, and whether β-arrestin1/2 hetero-oligomers and/or β-arrestin1 homo-oligomers per se drive tauopathy.

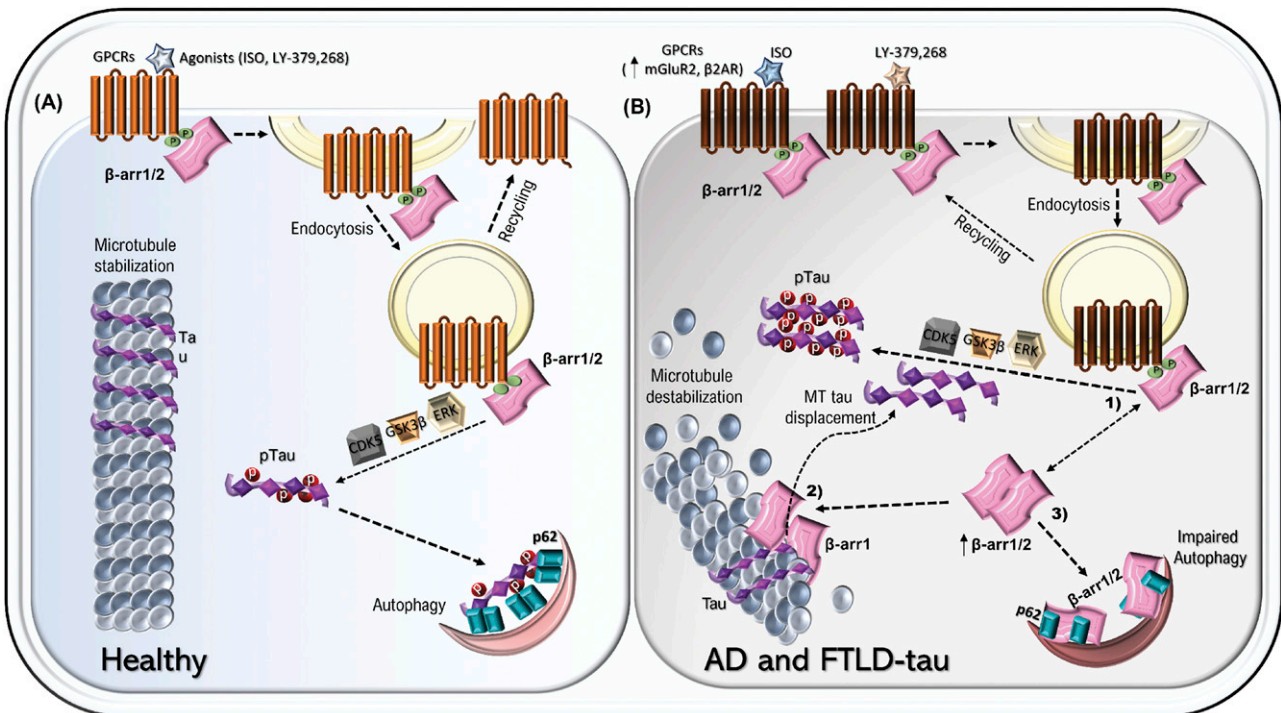

**Figure 7. Schematic model of β-arrestins in promotion of tauopathy.**
**(A)** In healthy brains, tau binds and stabilizes microtubules. β-arrestins mediate the desensitization and internalization of ligand-stimulated G protein–coupled receptors (GPCRs). This activity is also required for ligand-induced tau phosphorylation via kinases ERK, GSK3β, and CDK5. However, the autophagy cargo receptor p62/SQSTM1 readily binds and clears hyperphosphorylated tau through the autophagy-lysosome pathway. **(B)** In FTLD-tau and Alzheimer's disease brains, (1) GPCRs (β2AR & mGluR2) and β-arrestins1/2 are up-regulated, which drive tau phosphorylation upon ligand-induced stimulation. Also, (2) increased β-arrestin1 promotes tau dissociation from microtubules, thereby simultaneously disrupting microtubule stability and enabling tau missorting and phosphorylation. Furthermore, (3) β-arrestins1/2 inhibit p62-mediated clearance of pathogenic tau by blocking p62 body formation. Hence, the combined effects of β-arrestin1 in GPCR ligand-stimulated tau phosphorylation, tau displacement/missorting, microtubule disruption, and impaired p62-mediated tau clearance ultimately drive tauopathy in Alzheimer's disease and FTLD-tau.

In addition to these activities uncovered in this study, β-arrestin1 also promotes Aβ production and deposition in vivo (Liu et al, 2013). Hence, targeting β-arrestin1 represents a promising point of therapeutic intervention that can simultaneously alleviate Aβ and tau pathogenesis. Although *Arrb1*$^{-/-}$ mice exhibit impaired desensitization to β-adrenergic receptor stimulation in the heart, they are grossly normal, fertile, and do not display any physical or behavioral abnormalities (Conner et al, 1997). Therefore, reducing β-arrestin1 level or activity could be beneficial strategies to mitigate tauopathies including AD.

# Materials and Methods

### Patient samples

Frontal cortex tissue samples of pathologically confirmed FTLD-tau and age-matched nondemented controls were obtained from Emory ADRC (P50 AG025688).

### Animal models

The following mouse strains were used in this study: WT C57BL/6, PS19, and *Arrb1*$^{+/-}$ mice. C57BL/6J (Jackson Laboratory line 000664),

PS19 (Jackson Laboratory line 008169), and *Arrb1*$^{-/-}$ (Jackson Laboratory line 011131) were all obtained from Jackson laboratory. *Arrb1*$^{-/-}$ mice (AbdAlla et al, 2009a) and PS19 (Yoshiyama et al, 2007) mice have been characterized. Mice were housed under pathogen-free conditions, and all experiments involving mice were performed in accordance with approved protocols by the Institutional Animal Care and Use Committee (IACUC) at the University of South Florida.

### Primary neuronal cultures

Primary neurons were obtained from postnatal day 0 mice. Cortex and hippocampus were dissected in cold HBSS and digested with 0.25% trypsin. Neurons were plated on poly-D-lysine–coated plates or coverslips and maintained in neurobasal media with Glutamax and B27 supplement as previously described (Woo et al, 2017a).

### DNA transfection, constructs, and siRNA

pcDNA3 β-arrestin1-HA (14693; Addgene) (Luttrell et al, 1999), pcDNA3 β-arrestin1-Flag (14687; Addgene) (Luttrell et al, 1999), and pMXs-puro GFP-p62 (38277; Addgene) (Itakura & Mizushima, 2011) were obtained from Addgene. β-arrestin1-GFP and β-arrestin 1-myc constructs were kind gifts from Dr. Robert Lefkowitz at Duke

University. β-arresitn1 ON-TARGET plus SMART pool siRNA was purchased from Horizon Discovery. DNA constructs and siRNAs were transiently transfected with Lipofectamine 2000 and Opti-MEM.

## Recombinant proteins

pFast-tau-his and pFast-β-arrestin1-his constructs were transformed into DH10Bac-competent cells. After blue–white screening, DH10Bac strains were chosen to express recombinant Bacmids. Sf9 insect cells transfected with Bacmid were cultured for 3 d with Sf900 II SFM medium, then P1 generation virus in medium was collected and added to new Sf9 cells. After 2 d culture, Sf9 cells were harvested and lysed with lysis buffer (Tris 20 mM, pH 7.4, NaCl 150 mM, Triton X-100 1%, and 10 mM imidazole, with protease inhibitors). After centrifugation at 12,000$g$ for 15 min, supernatant was collected and shaken for 1 h at 4°C with GE Healthcare Ni Sepharose. Bound proteins on sepharose were washed three times with ice-cold lysis buffer, and recombinant proteins were eluted with ice-cold elution buffer (Tris 20 mM, NaCl 150 mM, and 200 mM imidazole), after which proteins were dialyzed in dialysis buffer (Tris 20 mM, NaCl 150 mM, and DTT 1 mM) at 4°C overnight.

## SDS–PAGE and Western blotting

Mouse brain extracts or cells were lysed in RIPA buffer (1% NP-40, 0.1% sodium dodecyl sulfate, 50 mM Tris pH 7.4, 150 mM NaCl, and 2 mM ethylenediaminetetraacetic acid) with protease and phosphatase inhibitors. After equalizing protein concentration with BCA assay, lysates were mixed with 4X LDS sample buffer and loaded on SDS–PAGE gels. Membranes were blocked with 5% milk in TBS-T for 1 h at room temperature. After blocking, membranes were probed with indicated primary antibodies for overnight at 4°C and incubated with horseradish peroxidase–linked secondary antibodies for 2–4 h at room temperature.

## Immunoprecipitation

Cells were lysed with CHAPS buffer (30 mM Tris-Cl, pH 7.5, 150 mM NaCl, and 1% CHAPS) with protease and phosphatase inhibitors. After equalizing protein concentration, lysates were preincubated with IgG beads for 1 h and washed with CHAPS buffer. Lysates were incubated with indicated primary antibody with IgG beads for overnight at 4°C, and proteins were eluted with 4X LDS sample buffer with boiling for subsequent SDS–PAGE and Western blotting.

## Sarkosyl-insoluble and soluble extraction

Sarkosyl extraction was performed as previously described (Woo et al, 2019). Briefly, brain homogenates were lysed with A68 buffer containing 10 mM Tris–HCl, pH 7.4, 0.8 M NaCl, 10% sucrose, and 1 mM EGTA. Samples were centrifuged at 400$g$ for 20 min at 4°C. After a centrifugation, 1% sarkosyl was added to the supernatants. The samples were incubated for 1 h and 30 min and centrifuged at 80,000$g$ for 30 min at room temperature. The pellets were resuspended in 100 $\mu$l of 50 mM Tris–HCl, pH 7.4 and subjected to SDS–PAGE.

## Tubulin polymerization assay

Tubulin polymerization was measured by the absorbance readings at 340 nm using the tubulin polymerization assay kit (Cytoskeleton Inc.). The concentration of tubulin was 3 mg/ml in 0.5 mM EGTA, 2 mM MgCl$_2$, 1 mM GTP, 80 mM PIPES, pH 6.9, and total polymerization volumes were 100 $\mu$l.

## Microtubule-binding assay

Microtubule-binding assay was performed by microtubule-binding protein spin-down assay kit (Cytoskeleton Inc.). Stable microtubules between 5 and 10 $\mu$m in length were used for the assay. After incubating stable microtubules with recombinant proteins, microtubule-associated proteins were pulled down at 100,000$g$.

## Immunofluorescence

Cells were washed with ice-cold PBS and fixed with 4% paraformaldehyde at room temperature. After fixation, cells were washed with 0.2% triton in TBS. Mice were perfused with PBS and fixed with 4% paraformaldehyde. 25-$\mu$m sections were washed with 0.2% triton in TBS. After washing, the cells and tissue sections were blocked with 3% normal goat serum with 0.1% Triton X-100 for 1 h at room temperature and incubated with indicated primary antibodies overnight at 4°C. After washing with PBS three times, cells and tissue sections were incubated with secondary antibodies for 45 min at room temperature. Images were obtained with the Olympus FV10i confocal microscope (Tokyo, Japan) or Zeiss LSM880 confocal microscope (Germany). Immunoreactivities were quantified using the Image J software (National Institutes of Health). All comparison images were acquired with identical laser intensity and exposure time. Investigators were blinded to experimental conditions during image acquisition and quantification.

## Generation of β-arrestin1-shRNA lentivirus

β-arrestin1-shRNA plasmid was obtained from Abm. Lentivirus vectors were co-transfected with pVSVG and Pax2 using polyethylenimine (PEI) in HEK293 cells for overnight. The medium was removed and replaced with serum-free medium the next day. After 72 h of incubation, the medium was collected and centrifuged to remove cell debris. Virus was filtered through a syringe filter (0.2–0.45 $\mu$m).

## Electrophysiology

Electrophysiological recording was performed as we previously described (Woo et al, 2017b, 2019). Briefly, brain slices were prepared from 4-mo-old WT, PS19, and PS19/*Arrb*1$^{+/-}$ mice. Input/output (IO) curve, PPF, and LTP were measured.

## Quantitative real-time RT-PCR

Quantitative real-time RT-PCR was performed using either Roche LightCycler 96 System (Life Science) or QuantStudio 3 Real-Time

PCR Systems (Thermo Fisher Scientific). After isolating total RNA using Trizol reagent (Invitrogen), total RNA was reverse transcribed and subjected to quantitative PCR analysis with SYBR Green master mix (Invitrogen) or Brilliant III SYBR Green qRT-PCR single-step master mix (600886-51; Life Technologies) was used for quantitative PCR analysis. The comparative threshold cycle (Ct) value was used to calculate the amplification factor, and the relative amount of $\beta$-arrestin1 or tau was normalized to GAPDH. The primer sequences: Human Tau-forward 5′-CCAAGCTCGCATGGTCAGTA-3′ and reverse-5′-GGCAGACACCTCGTCAGCTA-3′; human $\beta$-arrestin1 forwards: 5′-TGGAGAACCCATCAGCGTCAAC-3′ and reverse-5′ AGGCAGATGTCTGCATACTGGC-3′; human GAPDH-forward 5′-AAGGTCGGAGTCAACGGATT-3′ and reverse-5′-CCATGGGTGGAATCATATTGG-3′.

### Morris water maze

The mice were individually housed and handled for a minimum of 2 min 1 wk before the MWM test. MWM was performed as previously described (Morris, 1981). Briefly, the pool (120 cm diameter) was filled with water and non-toxic tempera white paint to make the water opaque. A hidden platform was placed 1 cm under the surface, and four signs with different colors and shapes were posted on the wall in each quadrant. The training period was 60 s trials with 1-h intervals for 4 sequential days. Each day of the 4-d training period, the mice were placed at an intersection at each quadrant with the order being randomly assigned that day. After the last day of training, the mice were probed at 24 h. During the probe, the hidden platform was removed, and the mice's activity was measured for 60 s. Mice behaviors were recorded using video tracking software (ANY-Maze). Experimenters were blind to genotype during trials.

### Statistical analysis and data presentation

Statistical analyses were performed by the GraphPad Prism 7.0 software (GraphPad Software) using paired or unpaired $t$ tests, and one- or two-way ANOVA with indicated post hoc tests. Data are shown as representative experiments. Box and whisker plots represent all data points with mean ± SEM. $P < 0.05$ was considered statistically significant.

## Data Availability

This study includes no data deposited in external repositories. The data that support the findings of this study are available from the corresponding authors on reasonable request. Further information and requests for reagents may be directed to and will be fulfilled by the corresponding author Dr. J-AA Woo (jaw330@case.edu).

### Ethics approval

IACUC and Institutional Biosafety Committees at University of South Florida and Case Western Reserve University approved that all the methods used in this study were performed in accordance with the relevant guidelines and regulations.

## Supplementary Information

## Acknowledgements

We thank Dr. Robert Lefkowitz at Duke University for providing $\beta$-arrestin1 constructs. We also thank Dr. Tamotsu Yoshimori and Noboru Mizushima for providing LC3 and p62 constructs. We thank Drs. Laura Blair and Chad Dickey for providing us with HeLa-V5-tau cell lines. We Thank Drs. Allan Levey and Marla Gearing for providing us with patients' brain samples. Patient brain samples were obtained from Emory ADRC center (P50 AG025688). This work was supported in part by grants from the National Institutes of Health (NIH) (R01AG059721-01A1, J-AA Woo; 1R01AG053060-01A1, DE Kang; R01AG067741-01, DE Kang & J-AA Woo), Veterans Affairs (BX004680, DE Kang), and Florida Department of Health (8AZ29, DE Kang & J-AA Woo, 20A01, J-AA Woo & DE Kang).

### Author Contributions

J-AA Woo: conceptualization, resources, data curation, software, formal analysis, supervision, funding acquisition, validation, investigation, visualization, methodology, project administration, and writing—original draft, review, and editing.
Y Yan: validation, investigation, methodology, and writing—review and editing.
TR Kee: data curation, validation, investigation, methodology, and writing—review and editing.
S Cazzaro: data curation, validation, investigation, methodology, and writing—review and editing.
KC McGill Percy: data curation, validation, investigation, methodology, and writing—review and editing.
X Wang: data curation, validation, investigation, methodology, and writing—review and editing.
T Liu: data curation, validation, visualization, methodology, and writing—review and editing.
SB Liggett: resources, validation, investigation, and writing—review and editing.
DE Kang: conceptualization, resources, funding acquisition, project administration, and writing—original draft, review, and editing.

### Conflict of Interest Statement

The authors declare that they have no conflict of interest.

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
