## [Reviewer comments · Life Science Alliance]

Life Science Alliance

β -arrestin1 promotes tauopathy by transducing GPCR signaling, disrupting microtubules and autophagy

Jung-A Woo, Yan Yan, Teresa Kee, Sara Cazzaro, Kyle McGill Percy, Xinming Wang, Tian Liu, Stephen Liggett, and David Kang

DOI: <https://doi.org/10.26508/lsa.202101183>

Corresponding author(s): Jung-A Woo, Case Western University, School of Medicine

Review Timeline:

Submission Date:	2021-08-04
Editorial Decision:	2021-09-09
Revision Received:	2021-11-15
Editorial Decision:	2021-11-17
Revision Received:	2021-11-18
Accepted:	2021-11-19

Transaction Report:

September 9, 2021

Re: Life Science Alliance manuscript #LSA-2021-01183-T

Prof. Jung-A A Woo
Case Western University, School of Medicine
Pathology
Cleveland, OH 44106

Dear Dr. Woo,

Thank you for submitting your manuscript entitled " β -arrestin1 promotes tauopathy by transducing GPCR signaling, disrupting microtubule and autophagy" to Life Science Alliance. The manuscript was assessed by expert reviewers, whose comments are appended to this letter. We invite you to submit a revised manuscript addressing the Reviewer comments.

Thank you for this interesting contribution to Life Science Alliance. We are looking forward to receiving your revised manuscript.

Sincerely,

B. MANUSCRIPT ORGANIZATION AND FORMATTING:

Reviewer #1 (Comments to the Authors (Required)):

Comments for β -arrestin1 promotes tauopathy by transducing GPCR-signaling, disrupting microtubule and autophagy

Scaffolding proteins β -arrestins (β -arrestin 1/2; ARRBs) regulate G-protein-coupled receptor (GPCR) desensitization and endocytosis. Increased expression of GPCR adaptor proteins have been reported in pathological conditions including Alzheimer's and Parkinson's disease. Previously, Woo et al. demonstrated that β -arrestin2 is elevated in human brains with frontotemporal lobar degeneration (FTLD). Increased β -arrestin2 impaired tau clearance and promoted tau aggregation by alleviating p62/SQSTM1 function.

In recent study, Woo et al. extended the study on the mechanistic basis of β -arrestin1-mediated GPCR stimulation effects on tauopathy. The authors showed the three main findings including

1. β -arrestin1 induced the dissociation of tau from microtubules, and inhibited tau-induced microtubule assembly.
2. β -arrestin1 and 2 share a common mechanism to promote pathological tau aggregation by blocking p62 autophagy cargo receptor.
3. Restored synaptic dysfunction was found in PS19/Arrb1^{+/-} transgenic mice.

These findings are like their previous study that β -arrestin2 oligomers impair the clearance of pathological tau and increase tau aggregates. Therefore, there are a couple of questions below, if the authors can comment or speculate on these issues will be useful to the readers.

In β -arrestin1 knockdown condition, do β -arrestin2 proteins have compensation mechanism to maintain dissociation of tau-microtubule and impairment of clearance mechanisms? # Do β -arrestin1 and 2 function dependently?

It will be useful to the readers if the authors can comment or speculate on these issues.

2. Minor suggestions

- Please add the corresponding molecular weight to the western blot figures.
- Figure 2G include the colocalization analysis.

Reviewer #2 (Comments to the Authors (Required)):

The study " β -arrestin1 promotes tauopathy by transducing GPCR signaling, disrupting microtubule and autophagy" by Woo and colleagues is an exciting straight-forward manuscript. In this manuscript Woo and colleagues show that β -arrestins are important for β 2AR and mGluR2-mediated increase in pathogenic tau. They also provide evidence that β -arrestin1 levels are increased in brains of Frontotemporal lobar degeneration (FTLD-tau) patients. This increase of β -arrestin1 leads to the accumulation of pathogenic tau, while reduced ARRB1 alleviates tauopathy and rescues impaired synaptic plasticity and cognitive impairments in PS19 mice. The presented studies show that β -arrestin1 drives tauopathy by destabilizing microtubules and impede p62/SQSTM1 autophagy flux by interfering with p62 body formation, which promotes pathogenic tau accumulation.

The manuscript addresses an up-to-date research question and the data presented may also lead to potential therapeutic applications. Overall, the individual experiments are straightforward and yield interesting results that will certainly have a considerable impact in the field. The results support a role for β -arrestin1 in tauopathy. In general, the study is well executed and provides convincing data. A few issues need to be addressed by the investigators in order to improve the manuscript and strengthen the findings of their study:

- 1) The authors show more β -arrestin1 in FTLD samples on protein levels. What about mRNA levels? The mRNA data was shown in a previous paper for β -arrestin2 by this group. This is of particular interest since the authors claim that pathogenic tau accumulation might also upregulate β -arrestin1 through a yet unknown mechanism ("feed forward loop"). Therefore, it might be interesting to analyze mRNA?
- 2) The authors mention several times that a tauopathy hallmark is the missorting of tau from axonal to dendritic compartments. Can they see that for tau after β -arrestin1 overexpression?
- 3) There are some mistakes in genotype labeling (Fig. 4 F/G/H) PS19 Arrb1 ^{-/-} , must be Arrb1 ^{-/+}

Reviewer #3 (Comments to the Authors (Required)):

This paper makes a very valuable contribution to the understanding of AD/tauopathy-related mechanisms by establishing that β -arrestin-related mechanisms can influence fundamental aspects of tauopathy including the excess accumulation of tau and p-tau levels, the relationship of tau to pathological microtubule states and the impairment of tau autophagy. This work also demonstrates a clear linkage of β -arrestin function to β 2AR or mGluR2 receptors, another important area in the AD field. These are each active and important areas in the tauopathy field. This paper is exceptionally well written, experimental findings are confirmed with additional approaches, especially executing both up- and down-regulation of key mechanisms and assessing key endpoints in multiple ways. The figures are of high quality. The data interpretation is appropriate and the Discussion is helpful.

In the Introduction or Discussion it would be helpful to indicate whether there is evidence that endogenous signaling by β 2AR or mGluR2 contribute to tauopathy in any models. Have antagonists for these receptors been studied in tauopathy models? Is there genetic evidence of β -arrestin 1 or 2 in AD or other tauopathies?

In the first paragraph of the results section, prior studies are listed that lead to the HeLa cell work in the present study. It would be helpful to mention which cells were used in the prior studies and how adding HeLa cells is important.

Should a distinction be made between β -arrestin "mediating" the receptor effect on tau, versus the more conservative interpretation that the receptor effects are " β -arrestin dependent"?

Can the importance of looking at both RIPA-soluble and -insoluble be explained?

In the β -arrestin shRNA experiments affecting tau levels as shown by western blots, it would be useful to comment on which species, or whether multiple species of tau are affected. Also, in the various western blot assessments, are there any changes in the ratio of p-tau/tau and hence any element of mechanism affecting tau phosphorylation as well as tau levels?

While the relative lack of phenotype for β -arrestin1 $-/-$ mice is mentioned, were actual specific behavioral/cognitive assessments done in these studies?

Does β -arrestin bind to cofilin? Would lowering β -arrestin levels affect cofilin mechanisms?

Was the β -arrestin1 antibody specificity confirmed in the present or prior studies with KO or similar approaches?

REVIEWER 1

In β -arrestin1 knockdown condition, do β -arrestin2 proteins have a compensation mechanism to maintain dissociation of tau-microtubule and impairment of clearance mechanisms?

β -arrestin1 knockdown is sufficient to prevent the dissociation of tau from microtubule and impaired clearance of pathogenic tau. We have previously shown that β -arrestin2 reduction is sufficient to mitigate tauopathy and promote the clearance misfolded tau. Hence, it is likely that their effects are either additive or cooperative (further discussed in the next question).

Do β -arrestin1 and 2 functions dependently?

It will be useful to the readers if the authors can comment or speculate on these issues.

β -arrestin1 and β -arrestin2 share 78% protein sequence homology, thereby sharing multiple biological functions, including GPCR desensitization and internalization. Double knockout mice for both genes causes neonatal death in mice (Zhang, Liu et al., 2010), whereas single knockout for either β -arrestin1 (Conner, Mathier et al., 1997) or β -arrestin2 (Bohn, Lefkowitz et al., 1999) are normal, suggesting that β -arrestin1 and β -arrestin2 play additive roles in their shared biological functions. As such, β -arrestin1 and β -arrestin2 may act in an additive fashion to drive tauopathy. However, it is also recognized that the two β -arrestins have differences in their functional capacities. β -arrestin2 has 6-fold greater affinity for clathrin (Goodman *et al*, 1996) and is 10-fold more efficient at internalizing β 2AR (Kohout *et al*, 2001). A 100-fold greater amount of β -arrestin1 is required to reconstitute wild-type β 2AR internalization compared to β -arrestin2 in β -arrestin1/2 double-knockout mouse embryonic fibroblasts (Kohout *et al*, 2001). In contrast, the angiotensin II type 1A receptor appears to be more sensitive to β -arrestin1 (Kohout *et al*, 2001). Multiple studies have shown that β -arrestin1 and β -arrestin2 constitutively form homo- and hetero-oligomers (Milano *et al*, 2006; Storez *et al*, 2005), and these oligomers are unable to bind to GPCRs (Boullaran *et al*, 2007; Breitman *et al*, 2012; Dinculescu *et al*, 2002; Gurevich & Benovic, 1993; Gurevich *et al*, 1995; Hanson *et al*, 2007b; Kim *et al*, 2012; Palczewski *et al*, 1994; Pulvermuller *et al*, 2000; Vishnivetskiy *et al*, 2011; Zhuang *et al*, 2013). Recently, we showed that β -arrestin2 oligomers mediate tauopathy by impairing pathogenic tau clearance (Woo *et al*, 2020), and β -arrestin2 oligomerization mutants show dominant-negative 'anti-oligomer' effects *in vitro* and *in vivo*. However, further studies are necessary to determine whether β -arrestin1 oligomerization mutants also act as 'anti-oligomer' dominant-negative, and whether β -arrestin1/2 hetero-oligomers and/or β -arrestin1 homo-oligomers *per se* drive tauopathy. These points are included in the discussion section (page 18-19).

Minor suggestions

- Please add the corresponding molecular weight to the western blot figures.

Thank you. We have added the corresponding molecular weight for all western blots.

- Figure 2G include the colocalization analysis.

We have added the colocalization analysis. Please see the new figure 2H.

REVIEWER 2

1. The authors show more b-arrestin1 in FTLD samples on protein levels. What about mRNA levels? The mRNA data was shown in a previous paper for b-arrestin2 by this group. This is of particular interest since the authors claim that pathogenic tau accumulation might also upregulate β -arrestin1 through a yet unknown mechanism ("feed forward loop"). Therefore, it might be interesting to analyze mRNA.

We have performed qRT-PCR to measure mRNA levels of β -arrestin1 in the frontal cortex of FTLD-tau patients and healthy controls (n=11 healthy control, and n=10 FTLD-tau patients). We found no significant difference in mRNA levels of β -arrestin1 between FTLD-tau patients and healthy controls. Please see the new supplementary figure 2C.

2) The authors mention several times that a tauopathy hallmark is the missorting of tau from axonal to dendritic compartments. Can they see that for tau after b-arrestin1 overexpression?

We have added the confocal images showing decreased colocalization between MAP2 (a somatodendritic marker) and tau in β -arrestin1-shRNA transduced hippocampal primary neurons compared to control-shRNA transduced primary neurons. Please see the new supplementary figure 5.

3) There are some mistakes in genotype labeling (Fig. 4 F/G/H) PS19 Arrb1 $-/-$, must be Arrb1 $-/+$

We apologize for the error in the figure legend. The manuscript text is correct. The mice that we performed the behavior tests are PS19/Arrb1 $-/-$ and PS19 littermates (from PS19/Arrb1 $+/-$ crosses with Arrb1 $+/-$ mice). We have corrected the figure legend.

REVIEWER 3

1. In the Introduction or Discussion it would be helpful to indicate whether there is evidence that endogenous signaling by β 2AR or mGluR2 contribute to tauopathy in any models. Have antagonists for these receptors been studied in tauopathy models? Is there genetic evidence of β -arrestin 1 or 2 in AD or other tauopathies?

As we mentioned in the result (Page 5), The genetic reduction of β 2AR mitigates tauopathy in vivo (Wisely et al., 2014), and β 2AR agonist significantly increases amyloid plaques in APP^{swe}/PS1 Δ E9 mice (Ni et al., 2006). Furthermore, Propranolol, a non-selective β -adrenergic receptor antagonist has been shown to attenuate cognitive impairments in Tg2576 mice (Dobarro et al., 2013).

mGluR2 stimulation also significantly increases tau phosphorylation (Lee et al., 2009), and mGluR2/3 antagonist, LY-341,495 increases interstitial fluid (ISF) tau (Yamada, Holth et al., 2014). Now we have mentioned manuscripts in the page 5, and we have cited all manuscripts accordingly.

Regarding the genetic evidence, Jian et al., have shown that the genetic variation of *ARRB2* is associated with late-onset AD in Han Chinese population (Jiang et al., 2014) while *ARRB1* genetic variants have not been exploited in AD/tauopathies.

2. In the first paragraph of the results section, prior studies are listed that lead to the HeLa cell work in the present study. It would be helpful to mention which cells were used in the prior studies and how adding HeLa cells is important.

Dobarro et al (2013), Wisely et al (2014), Lee et al (2004) have used hippocampus and cortex tissues. Lee et al (2009) have used rat primary cortical neurons.

HeLa cells stably overexpressing wildtype 4R0N human tau cells have been used by multiple groups and we have previously shown that β -arrestin2 promotes tau aggregation in the same cell lines. We would like to emphasize that we have also used primary neurons and tauopathy animal models in this manuscript.

3. Should a distinction be made between β -arrestin "mediating" the receptor effect on tau, versus the more conservative interpretation that the receptor effects are " β -arrestin dependent"?

We are afraid that we do not quite understand the difference in nuance, but we have revised the language to state that receptor effects on tau are dependent on β -arrestins.

4. Can the importance of looking at both RIPA-soluble and insoluble be explained?

RIPA buffer (containing 0.1% SDS & 1% NP-40) generates two fractions which are RIPA soluble and RIPA insoluble. RIPA-insoluble fraction contains high molecular weight proteins (Bai B et al., 2012; Li Q et al., 2016; Ngoka et al., 2008), and many extracellular matrix proteins and cytoskeleton proteins are also found in RIPA-insoluble fraction (Ngoka et al., 2008). Furthermore, it has been shown that posttranslational modifications can move protein into the RIPA insoluble fraction (Janes, 2015), and RIPA-insoluble total tau and pS396/404 tau have been measured in human and mouse brains (Forest et al., 2013). Therefore, we tested both RIPA-soluble and insoluble fractions to evaluate a more complete protein solubility profile. Indeed, we have found that RIPA-insoluble beta arrestin1 and beta-arrestin2 (Woo et al., 2020) levels are significantly increased in human FTLD-tau patients compared to healthy controls.

5. In the β -arrestin shRNA experiments affecting tau levels as shown by western blots, it would be useful to comment on which species, or whether multiple species of tau are affected. Also, in the various western blot assessments, are there any changes in the ratio of p-tau/tau and hence any element of mechanism affecting tau phosphorylation as well as tau levels?

We used PS19 mice expressing 4R1N, and HeLa-V5-tau cells stably overexpressing human tau 4R0N, and we found that reduced β -arrestin1 significantly decreases tau levels in both settings. We did not measure the ratio of p-tau/tau because β -arrestin1 significantly alters total tau levels.

However, we did show that genetic reduction of *arrb1* reduces pS199/pS202-tau in brain by IHC.

6. While the relative lack of phenotype for β -arrestin1 $-/-$ mice is mentioned, were actual specific behavioral/cognitive assessments done in these studies?

As we mentioned in the discussion, β -arrestin1 knockout mice are viable and do not exhibit gross abnormalities (Conner et al, 1997). Previously, Liu et al. (Cell Res. 2013 Mar; 23(3): 351–365) have shown that reduced β -arrestin1 ameliorates A β pathology in APP/PS1 mice. In this study, the authors have used Morris water maze to assess the spatial learning and memory, and there were no significant differences in cognitive functions between WT and β -arrestin1 knockout (*Arrb1* $-/-$) mice.

7. Does β -arrestin bind to cofilin? Would lowering β -arrestin levels affect cofilin mechanisms?

Yes. As we mentioned in the discussion, β -arrestins do interact with cofilin, LIMK, and chronophin (Zoudilova et al., 2007, 2010). It has been shown that Protease-activated receptor-2 (PAR-2) mediated activation of cofilin requires β -arrestins to scaffold cofilin and chronophin, a cofilin phosphatase. (Zoudilova et al., 2007, 2010). Furthermore, β -arrestin2 translocates activated cofilin to dendritic spines to regulate NMDA-induced remodeling of dendritic spines, and primary neurons derived from *Arrb2* $-/-$ are resistant to A β -induced dendritic spine loss (Pontrello et al, 2012). Overactivation of cofilin has been implicated in AD, and it is possible that reduced β -arrestins could decrease active/non-phosphorylated cofilin.

8. was the β -arrestin1 antibody specificity confirmed in the present or prior studies with KO or similar approaches?

Yes. The β -arrestin1 antibody that we used is a monoclonal antibody (Cell signaling, D7Z3W). We have confirmed the specificity using cortical primary neurons transduced with control, β -arrestin1 shRNA or β -arrestin2 shRNA. While β -arrestin1 shRNA transduced primary neurons show markedly reduced β -arrestin1, β -arrestin2 shRNA transduced primary neurons do not show reduced β -arrestin1. Please see the new supplementary figure 3C.

November 17, 2021

RE: Life Science Alliance Manuscript #LSA-2021-01183-TR

Prof. Jung-A A Woo
Case Western University, School of Medicine
Pathology
2103 Cornell Rd
Cleveland, OH 44106

Dear Dr. Woo,

Thank you for submitting your revised manuscript entitled " β -arrestin1 promotes tauopathy by transducing GPCR signaling, disrupting microtubule and autophagy". We would be happy to publish your paper in Life Science Alliance pending final revisions necessary to meet our formatting guidelines.

- please add the Twitter handle of your host institute/organization as well as your own or/and one of the authors in our system
- please note that titles in the system and manuscript file must match
- please use the [10 author names, et al.] format in your references (i.e. limit the author names to the first 10)
- please add your main and supplementary figure legends to the main manuscript text after the references section;
- all figure legends should only appear in the main manuscript file, please remove them from the figures
- LSA allows supplementary figures, but no EV Figures; please update your callouts for the Supplementary Figures in the manuscript Fig EV1A=Fig S1A; while supplementary figures use the system supplementary Fig S1;
- please add callouts for Figure 7A-B to your main manuscript text;

FIGURE CHECKS:

- please provide original blots used for Figure 3E

A. FINAL FILES:

B. MANUSCRIPT ORGANIZATION AND FORMATTING:

Sincerely,

November 19, 2021

RE: Life Science Alliance Manuscript #LSA-2021-01183-TRR

Prof. Jung-A A Woo
Case Western University, School of Medicine
Pathology
2103 Cornell Rd
Cleveland, OH 44106

Dear Dr. Woo,

Thank you for submitting your Research Article entitled " β -arrestin1 promotes tauopathy by transducing GPCR signaling, disrupting microtubules and autophagy". It is a pleasure to let you know that your manuscript is now accepted for publication in Life Science Alliance. Congratulations on this interesting work.

DISTRIBUTION OF MATERIALS:

Again, congratulations on a very nice paper. I hope you found the review process to be constructive and are pleased with how the manuscript was handled editorially. We look forward to future exciting submissions from your lab.

Sincerely,
